# BMP signaling regulates dorsal skeletal growth in the sea urchin embryo

William B. Douglas and Charles A. Ettensohn*

## ABSTRACT

The development of the elaborate, calcified endoskeleton of sea urchin embryos is a model for understanding the dynamic nature of developmental gene regulatory networks and the control of biomineralization. While several signaling pathways have been shown to regulate gene expression and biomineral formation by sea urchin skeletogenic cells, important gaps in our understanding remain. Here, we focused on signals that regulate skeletogenesis along the dorsal-ventral axis of the late-stage embryo. We used a specific inhibitor of Type I BMP receptors, K02288, to show that BMP signaling regulates skeletal growth selectively in the dorsal region. K02288 treatment led to dorsal skeletal defects and inhibited the expression of genes typically expressed specifically in the dorsal skeletogenic cells, including biomineralization genes. Using RNA sequencing, we identified genes that were uniquely downstream of either the BMP or a ventral signaling pathway (the VEGF pathway) at late developmental stages and genes downstream of both pathways. Our findings establish BMP signaling as a key pathway regulating dorsal skeleton formation and show that BMP signaling functions in concert with VEGF signaling to define the dorsal-ventral axis of the skeleton.

KEY WORDS: Biomineralization, BMP, Sea urchin, Skeletogenesis, Syncytium, VEGF

## INTRODUCTION

Sea urchin skeletogenesis has been a valuable experimental model for studying cell fate specification, cell signaling, morphogenesis and biomineralization (McIntyre et al., 2014; Shashikant et al., 2018; Gildor et al., 2021). The cells that will produce the embryonic skeleton, the large micromeres, form at the 32-cell stage as a result of two, successive, unequal cell divisions (Summers et al., 1993). At the late blastula stage, the large micromere descendants undergo epithelial-to-mesenchymal transition (EMT) and enter the blastocoel, after which they are referred to as primary mesenchyme cells (PMCs). During gastrulation, the PMCs extend filopodia and migrate on the blastocoel wall. At the same time, these cells fuse with one another via their filopodia, creating a single, continuous syncytial network (Gibbins et al., 1969; Hodor and Ettensohn, 1998, 2008). As the PMCs migrate and fuse, they form two clusters on the ventrolateral surfaces of the blastocoel wall. These ventrolateral clusters are connected ventrally and dorsally by slender chains of PMCs. Mineral deposition begins at the mid-gastrula stage with the formation of a single, tri-radiate spicule rudiment in each ventrolateral cluster. The arms of the tri-radiate spicule rudiments subsequently elongate and branch in a stereotypical fashion resulting in the formation of additional skeletal elements, including the post-oral rods, anterolateral rods, body rods and other smaller elements.

A complex developmental gene regulatory network (dGRN) deployed in PMCs has been characterized extensively (Oliveri et al., 2008; Davidson and Peter, 2015; Martik et al., 2016; Shashikant et al., 2018; Gildor et al., 2021). This dGRN is regulated both temporally and spatially. The temporal control of the PMC dGRN can be divided into an early, cell-autonomous phase (Phase 1) and a second, signal-dependent phase (Phase 2). These two phases of dGRN regulation approximately correspond to the periods of development before and after formation of the PMC syncytium. Phase 1 is initiated by the enrichment of maternal β-catenin, Dishevelled and the Otx(α) transcription factor (TF) in the micromeres, which leads to the expression of the TF Pmar1 (Chuang et al., 1996; Emily-Fenouil et al., 1998; Logan et al., 1999; Oliveri et al., 2003; Weitzel et al., 2004; Peng and Wikramanayake, 2013). Pmar1 drives the expression of several TF-encoding genes, including the key skeletogenic regulators *alx1* and *ets1*, leading to activation of downstream components of the network (Kurokawa et al., 1999; Kitamura et al., 2002; Ettensohn et al., 2003; Revilla-i-Domingo et al., 2007; Oliveri et al., 2008; Sharma and Ettensohn, 2010; Rafiq et al., 2014; Khor et al., 2019).

During Phase 2, cell-autonomous regulation of the PMC dGRN ends, and skeletogenesis becomes dependent on external signals. This shift in regulation drives a major change in spatial patterns of gene expression from an early pattern, in which effector genes are expressed uniformly by all PMCs, to a late pattern, in which expression is restricted to localized, distal sites of active skeletal growth in both the dorsal and ventral regions (Harkey et al., 1992; Guss and Ettensohn, 1997; Cheers and Ettensohn, 2005; Sun and Ettensohn, 2014). Although PMCs are organized in a syncytium, the mobility of TFs and biomineralization proteins within the syncytium is limited, providing a mechanism for generating and maintaining non-uniform patterns of both gene expression and biomineral growth (Khor et al., 2023).

Several signaling pathways have been shown to influence skeletogenesis including the RTK, TGFβ, Wnt and FAK-ROCK-ERK pathways (Röttinger et al., 2008; Piacentino et al., 2015; Sun and Ettensohn, 2017; Thomas et al., 2023; Hijaze et al., 2024; Layous et al., 2025). Although the localized production of signaling ligands might explain localized gene expression patterns within the PMC syncytium, the relationship between most of these signals and the regulation of the PMC dGRN during Phase 2 is poorly understood.

Carnegie Mellon University, Department of Biological Sciences, Pittsburgh, PA 15213, USA.

*Author for correspondence (ettensohn@cmu.edu)

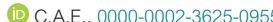 C.A.E., 0000-0002-3625-0955

One exception is the VEGF signaling pathway, which regulates gene expression in the ventral region of the PMC syncytium. The VEGF3 ligand is expressed by ectodermal cells overlying the ventrolateral clusters and is required for the formation of the tri-radiate spicule rudiments (Duloquin et al., 2007; Adomako-Ankomah and Ettensohn, 2013; Tarsis et al., 2022). The expression domain of VEGF3 then dynamically shifts as the ventral skeletal elements elongate, eventually becoming restricted to the ectoderm overlying the growing tips of the post-oral and anterolateral rods (Duloquin et al., 2007; Adomako-Ankomah and Ettensohn, 2013). Treatment of embryos with axitinib, a potent VEGFR inhibitor, inhibits the expression of biomineralization genes and skeletal growth on the ventral side of the embryo without affecting dorsal skeletogenesis (Sun and Ettensohn, 2014). This evidence strongly implies the presence of a second, unknown signal that is responsible for regulating the PMC dGRN in the dorsal region of the PMC syncytium (DS).

TGFβ signaling plays a key role in the formation of the dorsal-ventral axis during early sea urchin development (Duboc et al., 2004). Nodal and BMP2/4 are initially produced on the ventral side of the early embryo. The activity of BMP2/4 is inhibited ventrally by Chordin and the ligand spreads to the dorsal half of the embryo where it is essential for dorsal development (Lapraz et al., 2009). A combined morpholino knockdown of the Type I BMP receptors Alk1/2 and Alk3/6 leads to early dorsal defects and ventralization of the embryo (Haillot et al., 2015). Furthermore, Smad1/5/8, a downstream TF of the BMP signaling pathway, is preferentially phosphorylated on the dorsal side of the embryo beginning at the late blastula stage (Lapraz et al., 2009). Nodal/BMP signaling has been shown to contribute to left/right patterning of the coelomic pouches and anterior skeletal development during Phase 2 (Luo and Su, 2012; Molina et al., 2013; Su, 2014; Piacentino et al., 2015, 2016), but its possible role in dorsal skeletal patterning has not been explored. Notably, pSmad1/5/8 is enriched in the DS during Phase 2 of skeletogenesis (Haillot et al., 2015). These data demonstrate an early contribution of BMP signaling to general dorsal development and provide hints that this pathway might also regulate skeletogenesis in the DS during Phase 2.

In this study, we show that BMP signaling plays an essential role in regulating the PMC dGRN and skeletal growth in the DS. To elucidate the function of BMP signaling during Phase 2, we employed a small molecule inhibitor, K02288, that specifically inhibits the functions of Alk1/2 and Alk3/6 (Kerr et al., 2015). We analyzed the resulting changes in gene expression by hybridization chain reaction (HCR) *in situ* hybridization and RNA sequencing (RNA-seq). To better understand BMP signaling in the sea urchin during late embryogenesis and its role in dorsal skeletogenesis, we also carried out a comprehensive analysis of the spatiotemporal expression patterns of major components of the BMP signaling pathway at post-blastula stages. Our work shows that BMP signaling plays an essential role in dorsal skeletogenesis during Phase 2 and provides a foundation for future exploration of the mechanisms by which this pathway controls the PMC dGRN.

## RESULTS
### Dynamics of Smad1/5/8 activation
It has previously been observed that pSmad1/5/8 is initially concentrated broadly on the dorsal side of the embryo prior to PMC ingression and becomes restricted to the DS by the late gastrula stage (Lapraz et al., 2009; Chen et al., 2011; Haillot et al., 2015). We sought to confirm and extend these observations by analyzing the distribution of pSmad1/5/8 at later developmental stages, when the dorsal skeleton is forming. Using a pSmad1/5/8

monoclonal antibody, we monitored the spatial enrichment of pSmad1/5/8 from the mesenchyme blastula stage until the late pluteus stage at 5 days post-fertilization (dpf) in *Strongylocentrotus purpuratus*. pSmad1/5/8 immunostaining was only carried out on *S. purpuratus* as the pSmad1/5/8 antibody does not recognize the *Lytechinus variegatus* form of the protein, which has a slightly different sequence in the phosphorylated region. We found that, at the late mesenchyme blastula stage, pSmad1/5/8 was primarily enriched in the dorsal ectoderm and was faintly visible in the dorsal-most PMCs (Fig. 1A). By the early gastrula stage, pSmad1/5/8 levels were diminished in the ectoderm and elevated in the DS (Fig. 1B). A gradient of pSmad1/5/8 was apparent within the syncytium with pSmad1/5/8 levels diminishing towards the ventral side of the embryo. At later developmental stages, pSmad1/5/8 became further restricted in the ectoderm with only the small region overlying the dorsal PMC syncytium showing strong enrichment. This overall pattern of expression was maintained throughout later development and was still clearly present in late plutei (5 dpf) (Fig. 1C-G′). pSmad1/5/8 was also highly enriched in both coelomic pouches late in development (Fig. 1G′), as has been previously reported (Luo and Su, 2012). These observations indicate that BMP signaling is active in the DS during Phase 2.

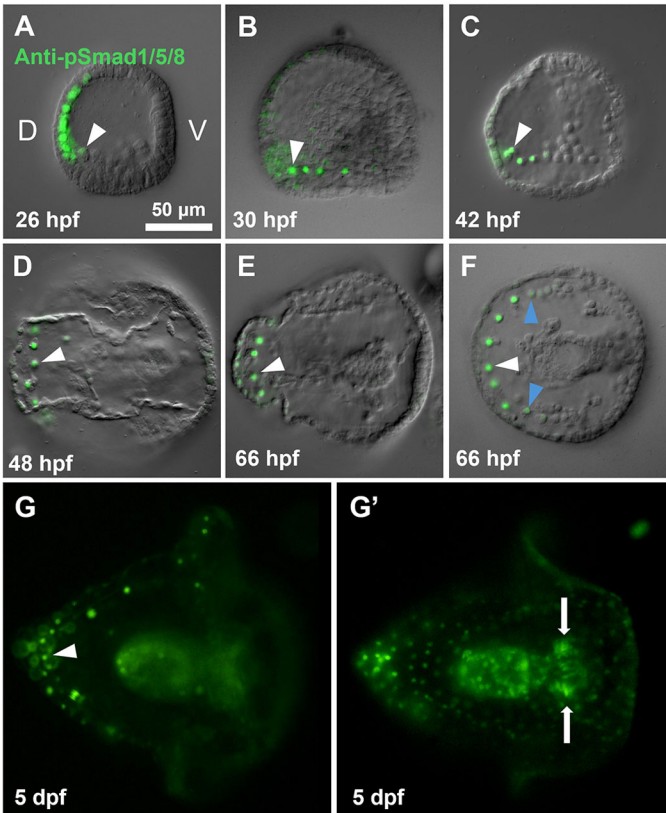

**Fig. 1. Smad1/5/8 activation is progressively restricted to dorsal PMCs.** (A-G′) *S. purpuratus* embryos were fixed and immunostained with anti-pSmad1/5/8 at the indicated times, from mesenchyme blastula to late pluteus stage. Embryos are oriented with dorsal sides on the left and ventral sides on the right. Enriched pSmad1/5/8 signal in dorsal PMCs is marked (white arrowheads). Diminishing pSmad1/5/8 in more ventral PMCs of the syncytium is also marked (F, blue arrowheads). A representative 5 dpf pluteus is shown at multiple focal planes to highlight the persistence of pSmad1/5/8 enrichment in the dorsal PMCs (G). pSmad1/5/8 is also strongly enriched in the coelomic pouches of pluteus stage embryos (white arrows) (G′). D, dorsal; V, ventral.

## BMP inhibitor validation

To test the role of BMP signaling specifically during Phase 2 without affecting early development, we employed a small molecule inhibitor, K02288, which selectively inhibits the Type I BMP receptors Alk1, Alk2, Alk3 and Alk6 (Kerr et al., 2015). Sea urchins have three Type I BMP receptors: Alk1/2, Alk3/6 and Alk4/5/7 (Lapraz et al., 2006). Alk4/5/7 plays an important role in anterolateral rod formation, but is not required for dorsal skeleton formation (Piacentino et al., 2015). Alk1/2 and Alk3/6 are necessary for receiving BMP2/4 input, and morpholino knockdown of Alk1/2 alone phenocopies BMP2/4 knockdowns, whereas knockdown of both receptors produces more extreme dorsal developmental defects (Haillot et al., 2015). Therefore, K02288 provides a means of specifically testing the role of Alk1/2 and/or Alk3/6 during Phase 2.

To evaluate the efficacy and specificity of K02288, we tested whether treatment with the drug phenocopied morpholino knockdown of the early dorsal BMP pathway. *L. variegatus* embryos were fertilized, immediately transferred to carrier (DMSO)

(Fig. 2A,A′) or drug (0.5 µM K02288) (Fig. 2B-C′) in artificial seawater (ASW), and cultured for 24 h. K02288-treated embryos produced supernumerary tri-radiate spicule-rudiments in a radialized pattern (32/36) (Fig. 2B-C′), a morphological phenotype reminiscent of Alk1/2+Alk3/6 double knockdowns seen in previous work (Haillot et al., 2015). Early embryos were particularly sensitive to K02288, requiring a lower concentration than later stages to survive. We also noted variability in the penetrance of the radialized phenotype between embryo batches, although in some batches this penetrance reached ∼90%.

We next tested whether K02288 effectively inhibited Smad1/5/8 activation. *S. purpuratus* embryos were transferred to carrier (DMSO) or drug (1 µM K02288) in ASW at the early gastrula stage [28 hours post-fertilization (hpf)], and cultured for an additional 20 h (early pluteus stage). The embryos were then fixed and immunostained with a pSmad1/5/8 antibody. DMSO-treated embryos showed strong pSmad1/5/8 enrichment in the DS (131/132) (Fig. 2D). In contrast, K02288-treated embryos showed a complete loss of pSmad1/5/8

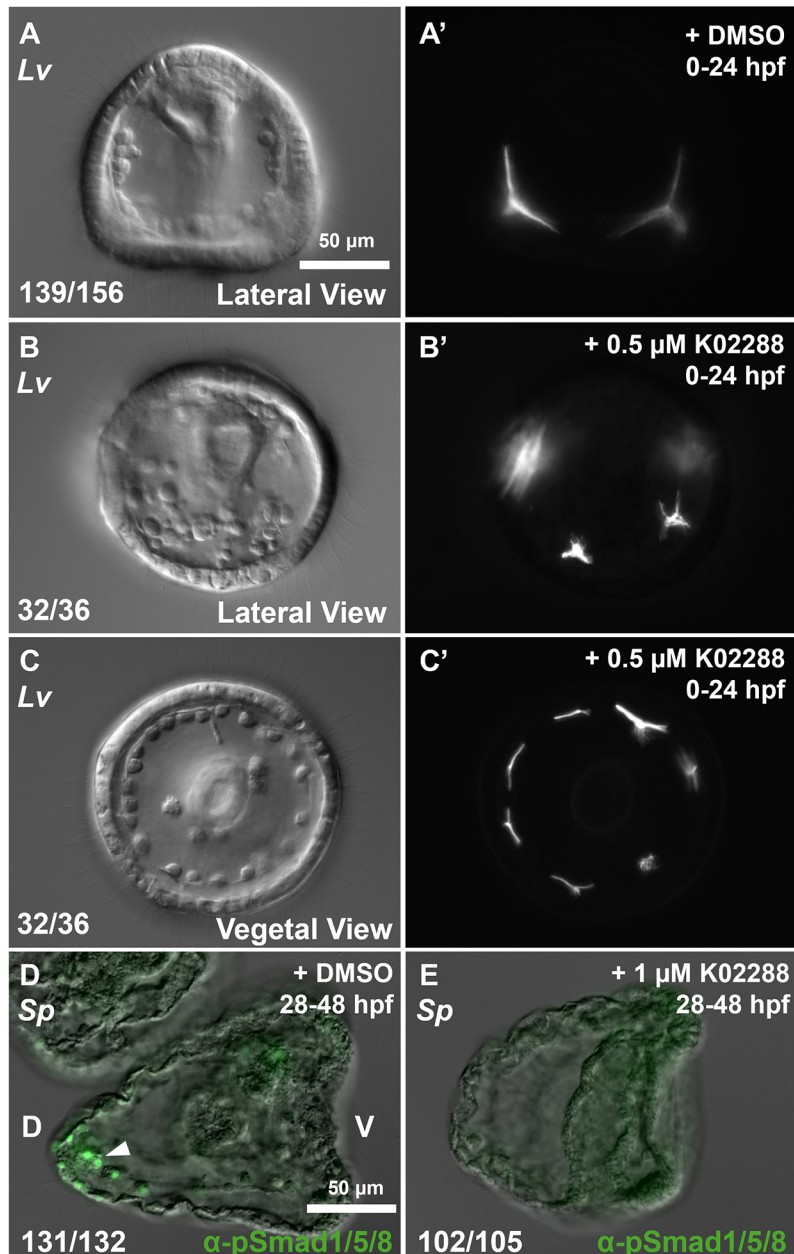

**Fig. 2. K02288 treatment phenocopies BMP pathway knockdowns and inhibits Smad1/5/8 phosphorylation.** (A-C′) Morphology of sibling *L. variegatus* embryos treated continuously from fertilization with DMSO (control) or 0.5 µM K02288. Morphologies were quantified at 24 hpf. Embryos were more sensitive to the drug at earlier stages and so a lower concentration of K02288 was used when treating immediately after fertilization. Images sharing the same letter show the same embryo under different optical conditions (DIC versus polarized light). (D,E) Effects of K02288 on the enrichment of pSmad1/5/8 in the DS (white arrowhead) was analyzed and pooled from three biological replicates (separate mating pairs). D and E are oriented with dorsal sides on the left and ventral sides on the right. Numbers at the bottom of images indicate the number of embryos exhibiting the morphology shown. D, dorsal; V, ventral.

staining in both the ectoderm and the DS (102/105) (Fig. 2D′). Embryos treated with K02288 during Phase 2 maintained a distinct dorsal-ventral axis and did not exhibit a radialized morphology. As pSmad1/5/8 normally accumulates in the dorsal-most PMCs by the early gastrula stage, these observations also suggest that continuous BMP signaling is necessary to maintain pSmad1/5/8 enrichment in the DS. Taken together, our findings indicate that K02288 is a potent and specific inhibitor of BMP signaling in sea urchin embryos.

### Effects of BMP signaling inhibition on skeletal growth

To test the function of BMP signaling during Phase 2 skeletogenesis, we treated embryos with K02288 continuously from the appearance of the tri-radiate spicule rudiments (mid-late gastrula stage) until the late pluteus stage (2 dpf) (Fig. 3A). BMP inhibition resulted in pluteus larvae with a distinctive morphology. K02288-treated larvae had splayed arms and a shortened scheitel compared to DMSO-treated siblings (Fig. 3B-C′). Most K02288-treated embryos (22/32) had visibly shortened body rods; often these rods were branched or distally curved giving them a hooked appearance (Fig. 3D,D′). We also noticed a particular morphological effect that K02288 had on the recurrent rods (RRs). In wild-type embryos, the RRs are made up of a proximal and distal segment with the transition between these two segments marked by a sharp bend. RRs in K02288-treated plutei typically failed to bend and lacked distal segments (27/32 embryos) (Fig. 3E,E′), and only a small percentage of embryos (5/32) had two normal recurrent rods. In some cases, one or both RRs were missing entirely in K02288-treated embryos (5/32), while only 1/34 DMSO-treated embryos lacked a single RR.

To quantify the effects of BMP inhibition on skeletal growth, we measured the lengths of the five major skeletal elements: anterolateral rods, RRs, post-oral rods, body rods, and ventral-transverse rods in DMSO-treated and K02288-treated embryos. The average length of almost all skeletal elements, with the exception of the ventral-transverse rods, was significantly shorter in K02288-treated embryos compared to DMSO treated-siblings (Fig. 3F). Notably, however, BMP inhibition showed the strongest impact on body rod growth based on the relative size of the five skeletal elements in K02288-treated embryos compared to DMSO-treated siblings (Fig. 3G). The average length of body rods in K02288-treated embryos was only 30% of the average length in DMSO-treated embryos whereas this proportion was >60% for all other elements. Because BMP inhibition had a preferential impact on body rod growth and RR branching, we concluded that this signaling pathway primarily regulates dorsal skeletogenesis.

### Spatiotemporal expression patterns of BMP ligands and receptors

Although the expression patterns of some BMP pathway components have been partially characterized (Duboc et al., 2004; Lapraz et al., 2006; Ben-Tabou de-Leon et al., 2013; Piacentino et al., 2015), there has been no comprehensive analysis of the developmental expression of BMP ligands and receptors during sea urchin embryogenesis. We used HCR in situ labeling to observe the spatiotemporal expression patterns of the Type I BMP receptors (alk1/2, alk3/6 and alk4/5/7), the single Type 2 BMP-specific receptor (bmpr2) and both BMP ligands (bmp2/4 and bmp5/8) in L. variegatus (Fig. S1). This analysis showed that alk3/6 and alk4/5/7 were weakly and ubiquitously expressed across multiple stages (Fig. S1) These mRNAs were detected in PMCs at low levels throughout embryogenesis, but were not enriched in any specific region of the PMC syncytium. In contrast, alk1/2 mRNA accumulated markedly in PMCs prior to invagination and was the first major component of the BMP signaling pathway activated in PMCs (Fig. 4A). This expression persisted during gastrulation, when alk1/2 mRNA was observed throughout the PMC syncytium (Fig. 4B). By the prism stage, alk1/2 expression was elevated in the dorsal chain of the PMC syncytium (Fig. 4C) and in PMCs at the tips of the post-oral arms (Fig. S1D′). At the pluteus larva stage, alk1/2 expression decreased in the DS but was still detectable there (Fig. 4D-E′).

The only known Type 2 receptor for BMP ligands in sea urchins is bmpr2 (Lapraz et al., 2006). We found that bmpr2 was weakly and ubiquitously expressed throughout gastrulation (Fig. 4F). bmpr2 expression then increased throughout the embryo by the prism stage, including in the DS (Fig. 4G). The widespread expression of bmpr2 persisted throughout later development and the mRNA remained detectable in the DS (Fig. 4H-I′).

The two BMP ligands, bmp2/4 and bmp5/8, showed very similar expression patterns within the skeletal syncytium (see also Luo and Su, 2012). At the late gastrula stage, only bmp2/4 was expressed at a detectable level in the PMC syncytium and was elevated in the dorsal region (Fig. 4J). By the prism stage, both bmp2/4 and bmp5/8 mRNAs were detectable and highly enriched in the DS, where they continued to be expressed throughout later development (Fig. 4K-Q, white arrowheads). Notably, at the early pluteus stage, both mRNAs were also expressed in the PMCs at the distal ends of the RRs and remained strongly enriched there at later developmental stages (Fig. 4L-Q′, black arrowheads). bmp2/4 appeared to be expressed in more cells along each RR than bmp5/8.

### HCR analysis of BMP-responsive genes in PMCs
#### TF-encoding genes

We next tested whether K02288 treatment inhibited the expression of genes that are normally selectively upregulated in the DS. During Phase 2, two TF-encoding genes, gata3 (also known as gataC) and scl, are selectively expressed in the DS of S. purpuratus (Solek et al., 2013). We observed that both genes were also expressed in the DS of L. variegatus and that gata3 mRNA was strongly enriched in the ventral midgut of late pluteus embryos (Fig. 5A,A′,D,D′). Notably, we observed that both gata3 and scl were also expressed in RR-PMCs of L. variegatus (Fig. 6A,A′,D,D′). Expression of gata3 in the RR-PMCs was strong and consistently present (53/59). scl expression in the RR-PMCs was observed less frequently (15/40), despite consistent expression in the dorsal region of the syncytium (39/40).

In the presence of K02288, the expression of both gata3 and scl was inhibited in the DS and in the RR-PMCs while expression of gata3 in the ventral midgut was unaffected (Figs 5, 6A-B′,D-E′). Overall, 27/42 embryos lacked detectable gata3 expression in both the DS and the RR-PMCs and 23/40 embryos lacked detectable scl expression in both the DS and the RR-PMCs. For cases in which expression was detectable, it appeared much fainter qualitatively compared to control embryos.

### BMP ligands – autoregulation of expression in the PMC syncytium

Previous work showed that bmp2/4 expression in the DS is lost in embryos treated from gastrulation with dorsomorphin, another Type I BMP receptor inhibitor (Luo and Su, 2012). We tested whether K02288 had the same effect on bmp2/4 and whether its expression in the RR-PMCs was similarly dependent on BMP signaling. We also tested whether bmp5/8 behaves similarly to bmp2/4 in response to the drug. We found that K02288 treatment strongly inhibited the expression of both mRNAs in the DS and

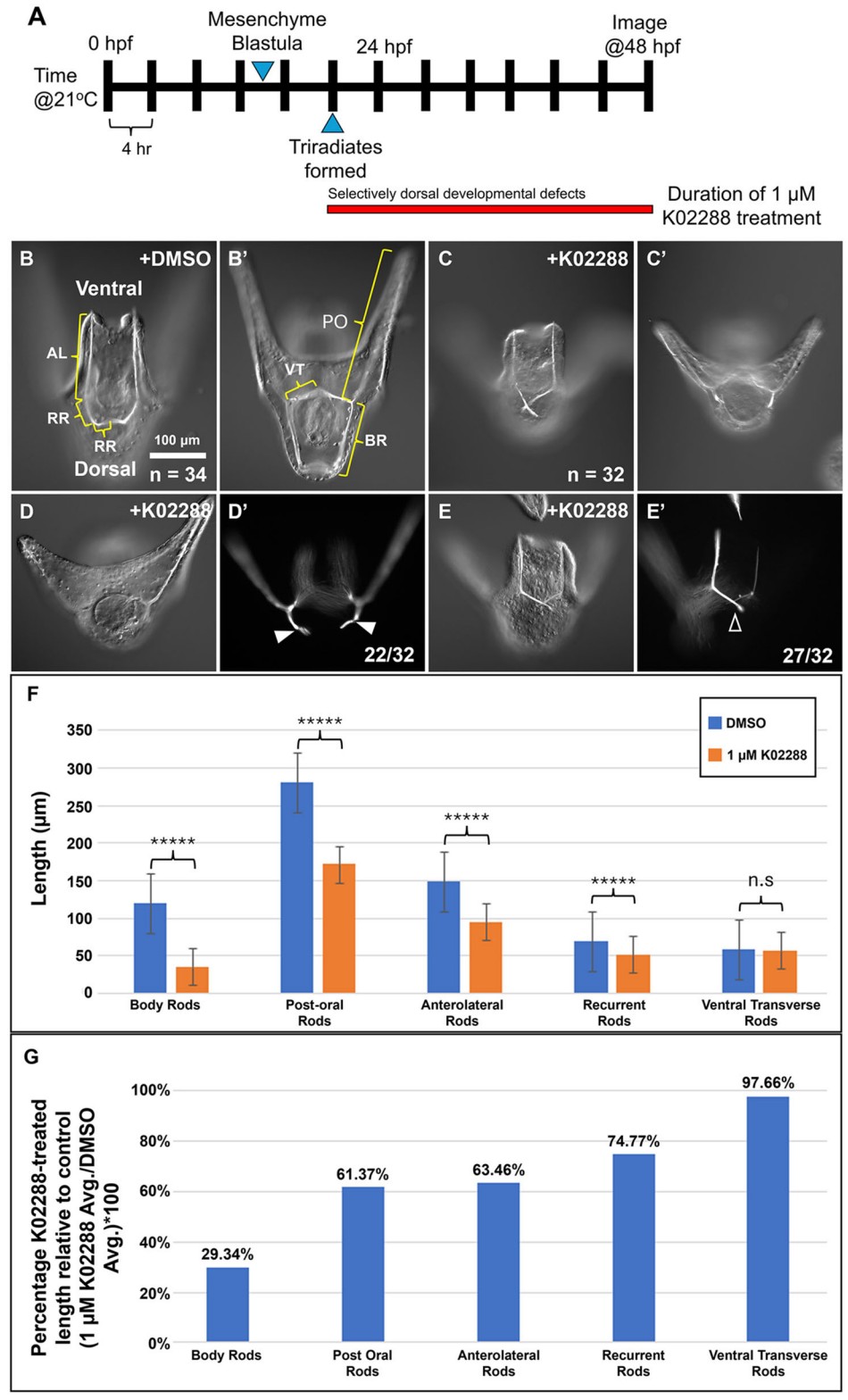

**Fig. 3. Inhibition of the BMP signaling pathway during Phase 2 of skeletogenesis selectively affects body rod and recurrent rod development.** (A) Timeline of treatment and embryo analysis. (B-C′) Representative images of DMSO-treated (34 embryos measured) and K02288-treated (32 embryos measured) embryos, showing the five skeletal elements that were measured: anterolateral rods (AL), recurrent rods (RR), post-oral rods (PO), ventral-transverse rods (VT) and body rods (BR). (D,D′) Representative K02288-treated embryo with a shortened body rod and a hooked scheitel (22/32) (white arrowheads). (E,E′) Representative K02288-treated embryo with perturbed recurrent rods that failed to bend and lack distal segments (27/32) (black arrowhead). (F) Average lengths for each skeletal element were compared between DMSO-treated and K02288-treated embryos by Student's *t*-test. (G) Proportional average length of each element in K02288-treated embryos compared to DMSO-treated embryos. Images sharing the same letter show the same embryo at different focal planes or under different optical conditions (DIC versus polarized light). Numbers at the bottom of images indicate the number of embryos exhibiting the morphology shown or the total number of embryos examined. n.s., not significant.

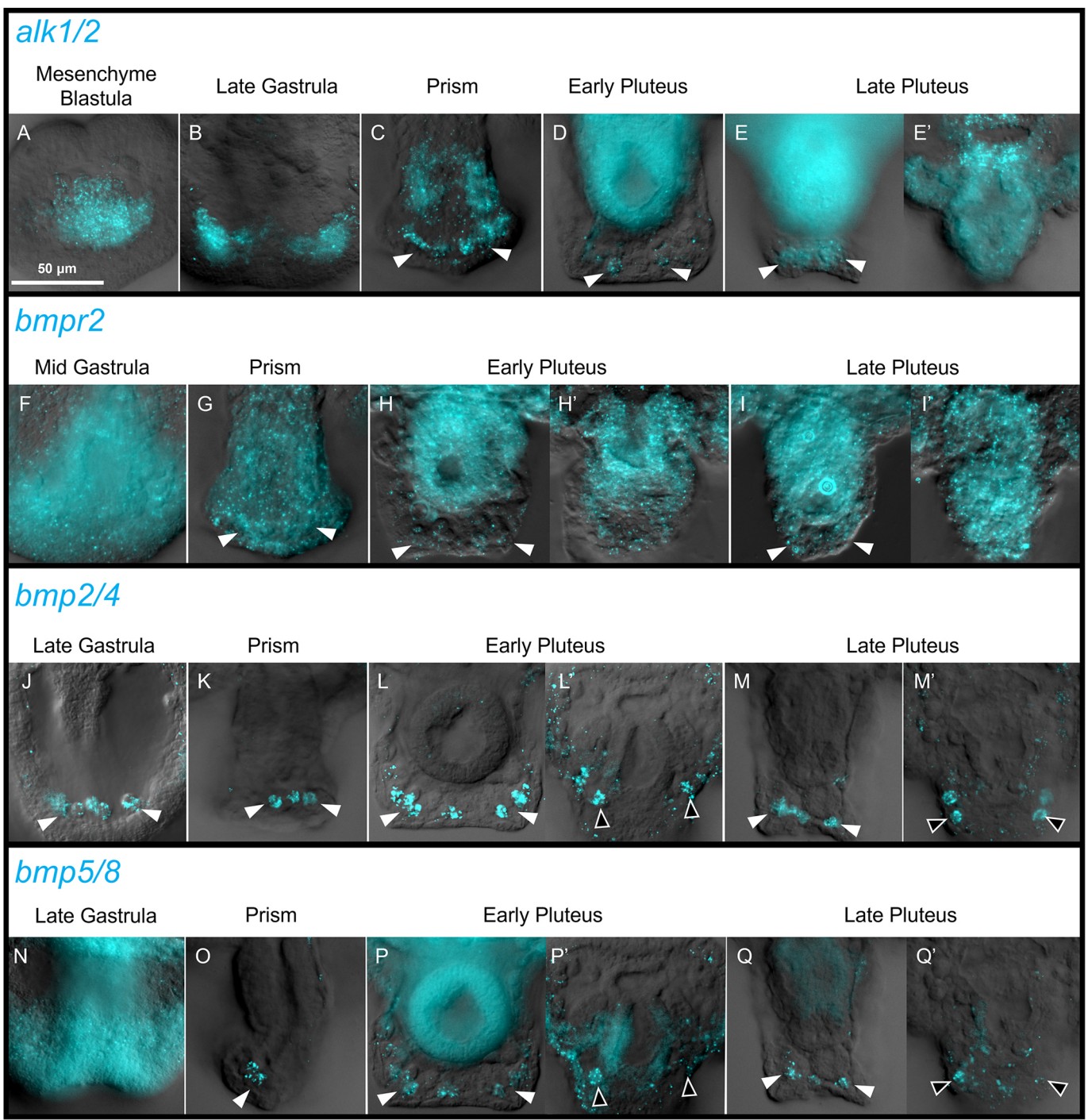

**Fig. 4. Components of the BMP signaling pathway are expressed in the DS and the distal RR-PMCs.** Fixed *L. variegatus* embryos from various stages were stained by HCR *in situ* hybridization. White arrowheads mark the DS when expression is present. Black arrowheads mark RR-PMCs when expression is present. The embryos show strong autofluorescence in the gut at later stages, which is exacerbated by the body being compressed on the slide. Some panels show the same embryo stained for multiple mRNAs and there are three such groups of images: C,G; L,L′,P,P′; M,M′,Q,Q′. Images of late gastrula embryos show lateral views. Prism and later stage embryos are oriented with the blastopore against the slide/coverslip such that the anterior-posterior axis is perpendicular to the image. Prism and later stage images show high magnification views of the dorsal scheitel (C-E,G,H,I,K,L,M,O,P,Q) or the recurrent rod PMCs (E′,H′,I′,L′,M′,P′,Q′). At least 30 embryos of each stage for each gene of interest were observed, and representative images showing expression patterns consistent with >90% of observed embryos are shown.

the RR-PMCs (Figs 5, 6G-H′,J-K′). Expression of *bmp2/4* was undetectable in both the DS and RR-PMCs of 26/40 embryos and was lacking in one region while faintly detectable in the other region in 5/40 embryos. *bmp5/8* expression was undetectable in both regions in 38/40 embryos.

We also observed that *bmp2/4* and *bmp5/8* were normally expressed in the ectoderm of the oral hood and along the ventral side of the post-oral arms (Fig. S2). Notably, K02288 treatment did not affect the expression of either mRNA in these regions, revealing a selective effect of BMP signaling on the expression of BMP ligands in PMCs.

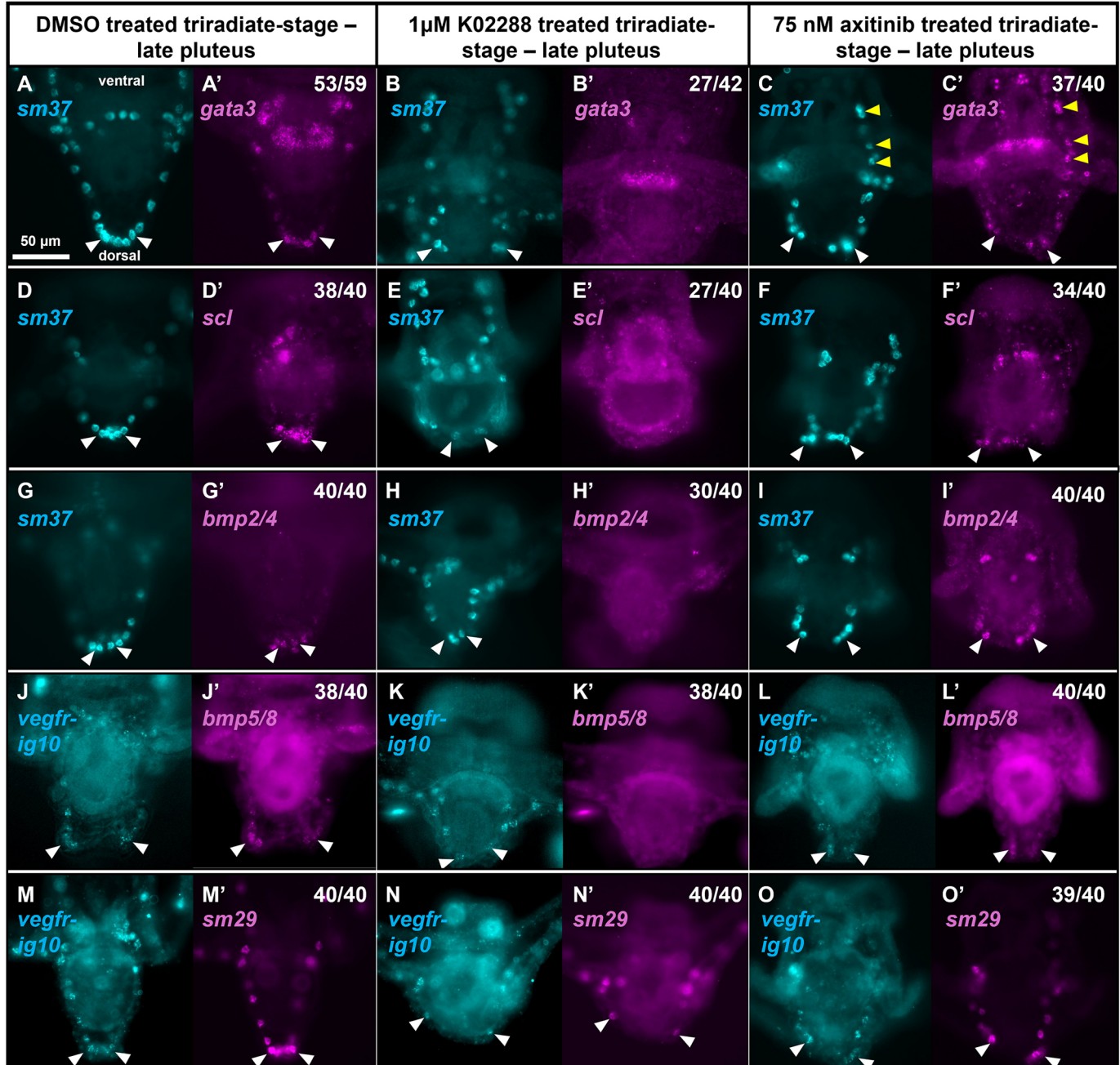

**Fig. 5. The expression of mRNAs enriched in the DS is dependent upon BMP signaling.** (A-O′) *L. variegatus* embryos were treated with DMSO, 1 µM K02288 or 75 nM axitinib continuously starting after the formation of the tri-radiate spicule rudiments. Embryos were fixed at the late pluteus stage (2 dpf) and the expression of various genes of interest was analyzed by HCR *in situ* hybridization (magenta). *sm37* and *vegfr-lg10* were used as counterstains to label the entire PMC syncytium (cyan). The number of DMSO-treated embryos with gene expression in the DS (marked with white arrowheads) is shown (A′,D′,G′,J′,M′). Representative images of K02288-treated embryos completely lacking detectable gene expression in the DS and the fraction of these embryos are shown (B′,E′,H′,K′). Representative images of axitinib-treated embryos with DS expression and the fraction of these embryos are shown (C′,F′,I′,L′,O′). In all cases (40/40), enriched expression of *sm29* in the DS is qualitatively reduced in K02288-treated embryos compared to control embryos and distal RR-PMC expression of *sm29* appears visually the same as other PMCs within the syncytium of K02288-treated embryos (N′). C′ shows the number of embryos with both DS expression and ectopic expression in ventral PMCs of the syncytium (yellow arrowheads). Images sharing the same letter(s) show the same embryo stained for different mRNAs.

## Biomineralization genes

Several different restricted patterns of gene expression have been observed within the PMC syncytium (Sun and Ettensohn, 2014). Using K02288, we sought to identify genes with DS-specific expression that was reliant on BMP signaling. In control embryos, *sm37* was expressed ubiquitously in the PMC syncytium.

Preliminary studies showed that K02288 had no effect on *sm37* expression; therefore, this mRNA was used as a marker for the entire PMC syncytium in drug-treated embryos (Figs 5, 6A-I′). We also observed the effect of K02288 on *sm29*. In control embryos, *sm29* mRNA was enriched in the dorsal region and RR-PMCs (Figs 5, 6M,M′). *sm29* was also expressed at lower levels in the remainder of

the syncytium except for the PMCs at the tips of the anterolateral arms, which lacked expression entirely (Fig. 6M,M′, red arrowheads). K02288 treatment reduced *sm29* expression within the DS and the RR-PMCs resulting in uniform levels of *sm29* expression throughout the syncytium in all observed cases (40/40) (Figs 5, 6N,N′).

## VEGF signaling is not required for expression of BMP signaling targets in the DS

It has been reported that the principal VEGF receptor, *vegfr-Ig10*, is expressed throughout the entire PMC syncytium albeit weakly in the DS compared to the strong expression at the growing arm tips (Duloquin et al., 2007). We observed the same pattern when using

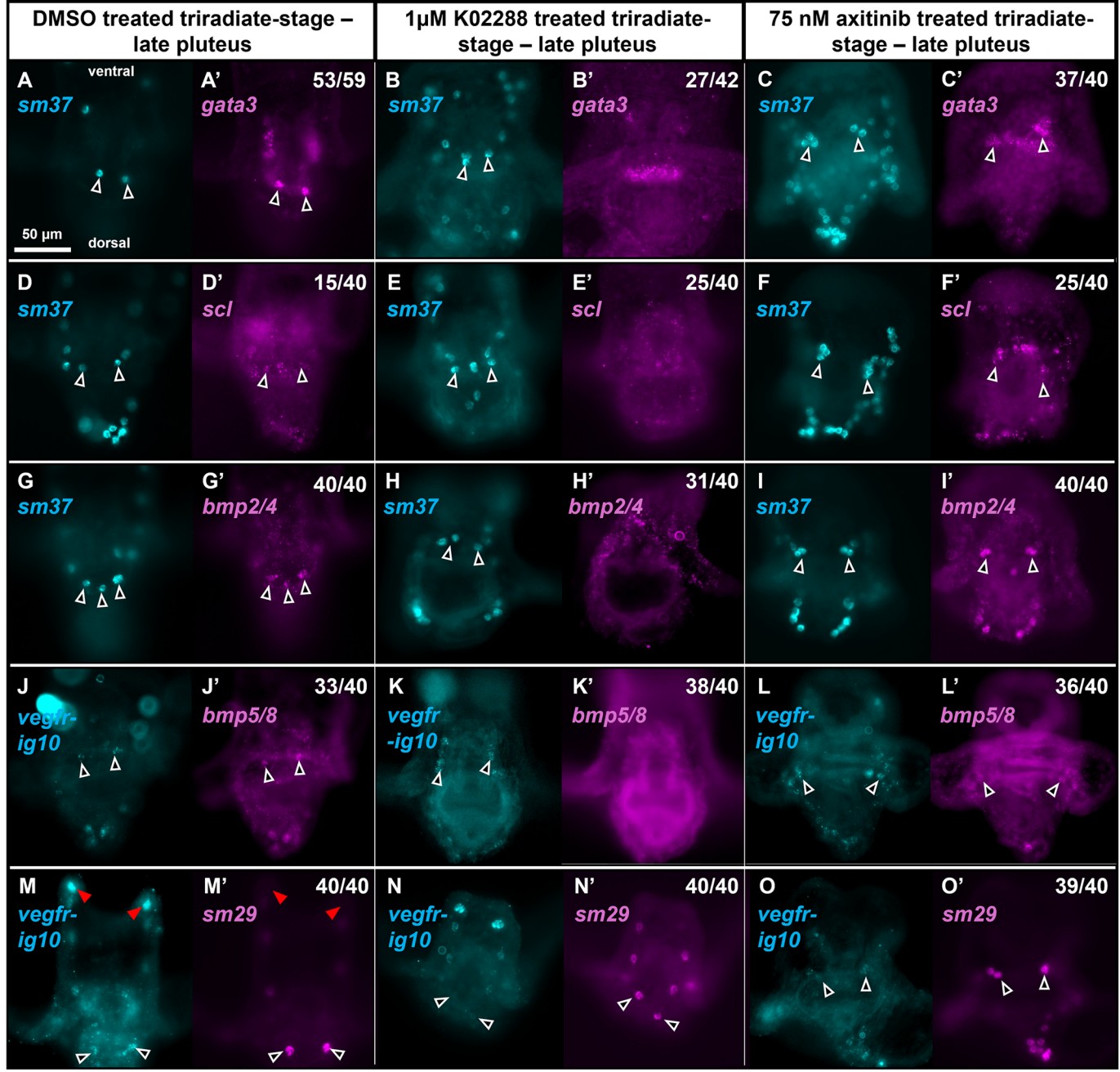

**Fig. 6. The expression of mRNAs enriched in the RR-PMCs of the syncytium is dependent upon BMP signaling.** (A-O′) *L. variegatus* embryos were treated, collected and stained as previously described in Fig. 5. The number of DMSO-treated embryos with gene expression in the RR-PMCs (marked with outline arrowheads) is shown (A′,D′,G′,J′,M′). Representative images of K02288-treated embryos completely lacking detectable gene expression in the RR-PMCs and the fraction of these embryos are shown (B′,E′,H′,K′). Representative images of axitinib-treated embryos with RR-PMC expression and the fraction of these embryos are shown (C′,F′,I′,L′,O′). In all cases (40/40), enriched expression of *sm29* in the distal most RR-PMCs is qualitatively reduced in K02288-treated embryos compared to control embryos and distal RR-PMC expression of *sm29* appears visually the same as other PMCs within the syncytium of K02288-treated embryos (N′). Distal anterolateral PMC positions showing lack of wild-type *sm29* expression are marked (red arrowheads) (M,M′). Images sharing the same letter(s) show the same embryo stained for different mRNAs. Additionally, several images show the same embryos as corresponding images in Fig. 5 at different focal planes or of the same focal plane to draw attention to the recurrent rods instead. These groups of images are as follows: B,B′ and Fig. 5B,B′ ; E,E′ and Fig. 5E,E′; F,F′ and Fig. 5F,F′; I,I′ and Fig. 5I,I′; and N,N′ and Fig. 5N,N′.

VEGFR as a PMC-specific marker in our HCR experiments (Figs 5, 6J,M). Although the *vegf3* ligand is not expressed dorsally, it has been previously reported that the *vegf2* ligand is expressed in dorsal PMCs (Kipryushina et al., 2013). Thus, we tested whether VEGF signaling was also necessary for the downstream targets of the BMP signaling pathway to be expressed in the DS and the RR-PMCs. Addition of axitinib, a small molecule inhibitor of VEGFR previously used in sea urchins (Adomako-Ankomah and Ettensohn, 2013), had no effect on the expression of any of the candidate genes in the DS or the RR-PMCs (Figs 5, 6). Surprisingly, *gata3* expression was consistently ectopically present in the ventral PMCs of the syncytium in axitinib-treated embryos (37/40) (Fig. 5C,C′). There was some ectopic, ventral expression within the PMC syncytium of *scl* and *bmp2/4*, but this pattern was not consistent between replicates.

### The BMP and VEGF signaling pathways show overlapping but distinct contributions to the regulation of the skeletogenic network

We next sought to determine the broader contribution of BMP-signaling to Phase 2 skeletogenesis. We performed bulk RNA-seq on embryos treated continuously with DMSO, 1 µM K02288 or 75 nM axitinib from the late gastrula/early prism stage, when the tri-radiate spicule rudiments had formed, until the late pluteus stage. By comparing the effects of the two inhibitors, we sought to identify the unique contributions of the BMP and VEGF pathways to skeletogenesis during Phase 2 skeletogenesis as well as possible shared effects.

Our initial analysis identified a total of 1077 genes that were significantly altered in expression (up or down) in embryos treated with axitinib or K02288 compared to controls ($P \leq 0.05$) (Tables S1-S4): 494 genes were significantly downregulated and 295 genes were significantly upregulated by axitinib treatment compared to controls, and 196 genes were significantly downregulated and 224 genes were significantly upregulated by K02288 treatment compared to controls. The above values include 132 genes that had significantly altered gene expression in response to both drugs (Table S5). Of these genes, 94 showed no significant difference between drug treatments, while 38 genes showed a significant difference between inhibitors (Table S6) either in the intensity of expression change (19/38) or by exhibiting opposite reactions to the inhibitors (19/38 were upregulated by one inhibitor and downregulated by the other compared to controls).

To focus specifically on genes expressed selectively by PMCs during Phase 2, we prepared a curated list of PMC-enriched genes expressed during late *L. variegatus* embryogenesis based on publicly available single-cell RNA-seq (scRNA-seq) data (Massri et al., 2021) and known gene expression patterns visualized by *in situ* hybridization (see Materials and Methods for additional details). A total set of 367 late PMC differentially expressed genes was identified this way in this manner the inhibitors (Tables S9, S10); 19 were sensitive to K02288 only, 30 were sensitive to axitinib only and 25 were sensitive to both drugs (Fig. 7A). Most of the genes affected by the inhibitors (65/74, or 88%) showed decreased expression in drug-treated embryos compared to controls (Tables 1-3). We supplemented this gene set with an additional 46 genes that were shown in previous studies to be expressed selectively by PMCs during late embryogenesis, including a set of 29 TF-encoding genes (Solek et al., 2013; Sun and Ettensohn, 2014; Valencia, 2018; Massri et al., 2021).

While only a small subset of PMC-specific genes was affected by K02288 (Fig. 7A), the affected genes encoded a variety of functional classes of proteins, several of which are known to play

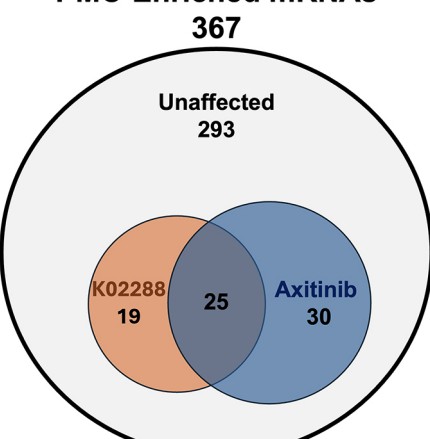

**PMC-Enriched mRNAs**
**367**

**Fig. 7. Venn diagram showing the proportion of curated PMC-enriched mRNAs with expression sensitive to K02288, axitinib or both drugs.** Approximately 20% (74/367) of the curated list of PMC-enriched mRNAs was affected by the drug treatments. The list of enriched mRNAs is based on 321 mRNAs identified as PMC enriched at late stages of development in previous single-cell RNA-seq work (Massri et al., 2021) and 46 manually curated genes based on known expression patterns (Solek et al., 2013; Valencia, 2018).

a role in skeletogenesis (Tables 1, 2). These included spicule matrix proteins, secreted matrix metalloproteinases and MSP130 family proteins. BMP signaling was also required for proper expression of *otop2L* and *p58b*, both known to be essential for skeletogenesis (Adomako-Ankomah and Ettensohn, 2011; Chang et al., 2021). Other K02288-sensitive genes encoded cytoskeleton remodelers, membrane-bound transporters, TFs (including *gata3*), and several proteins of unknown function (Tables 1, 2).

### DISCUSSION
The skeletogenic program deployed in PMCs provides a striking example of the dynamic nature of dGRNs. The initial deployment of the PMC dGRN (Phase 1) is controlled by molecular asymmetries present in the unfertilized egg and is independent of cell–cell signaling (Oliveri et al., 2008; Ettensohn, 2020; Kipryushina and Yakovlev, 2020; Molina and Lepage, 2020). This initial phase of dGRN deployment establishes the identity of the PMCs and is sufficient to support EMT and fusion but not overt skeletogenesis. A major shift in regulatory control occurs during gastrulation, when the PMC dGRN becomes responsive to external signals. The localized nature of ectoderm-derived signals leads to regional differences in gene expression and skeletal growth within the PMC syncytium (Harkey et al., 1992; Guss and Ettensohn, 1997; Adomako-Ankomah and Ettensohn, 2013; Sun and Ettensohn, 2014). Although PMCs are organized in a syncytium, the mobility of TFs and biomineralization proteins within the syncytium is limited, providing a mechanism for generating and maintaining non-uniform patterns of gene expression and biomineral growth (Khor et al., 2023). While the architecture of the PMC dGRN during Phase 1 has been studied intensively, we know much less about the dGRN during Phase 2, when the skeleton forms.

This work has identified the BMP signaling pathway as an important regulator of sea urchin skeletogenesis during Phase 2. Inhibition of BMP signaling led to significant skeletal growth defects on the dorsal side of the embryo. In addition, we observed by HCR *in situ* hybridization that the localized expression of genes in

**Table 1. PMC-enriched mRNAs downregulated specifically by K02288**

| Echinobase gene symbol | Lv LOC ID | Function |
|---|---|---|
| coro7[1] | LOC121407526 | Actin binding protein family, cytoskeleton remodeling, mobility |
| exoc4 | LOC121425692 | Exocyst Complex 4 (predicted) |
| LOC105437764 | LOC121432123 | Formin-like protein 14, actin polymerization (predicted) |
| gata3[2] | LOC121411816 | Gata binding factor, transcription factor, DNA binding, blastocoelar specification, mesodermal development, coelomic pouch development |
| LOC100889952 | LOC121430801 | Kelch family protein, cell remodeling, cytoskeleton remodeling, actin polymerization, skeletal muscle formation (predicted) |
| MMP16 (mmp16) | LOC121409096 | Matrix metalloproteinase 16, biomineralization (predicted) |
| LOC577128 (mmp24) | LOC121409095 | Matrix metalloproteinase 24, biomineralization (predicted) |
| LOC579173[3] (otop2L) | LOC121426418 | Proton channel, biomineralization, otopetrin 2-like |
| zeb2 | LOC121411537 | Smad Interacting Protein, co-effector/repressor, DNA binding (predicted) |
| sm30f[4] | LOC121406359 | Spicule matrix protein, biomineralization |
| sm29[5] | LOC121405931 | Spicule matrix protein, biomineralization |
| LOC754831 | LOC121416238 | Transient receptor potential cation channel (predicted) |
| LOC105439232 | LOC121418149 | Type II membrane bound glycoprotein, enzymatic functions (predicted) |
| LOC578886 | LOC121427246 | Tyrosine-protein phosphatase non-receptor type 11 (predicted) |
| LOC115919418 | LOC121411291 | Unknown |
| LOC105443651 | LOC121418355 | Unknown |
| Uncharacterized | LOC121413455 | Unknown, does not align well to any S. purpuratus genes, annotated as ncRNA on NCBI |

[1]Terasaki et al., 1997; [2]Solek et al., 2013; [3]Chang et al., 2021; [4]Wilt et al., 2013; [5]Illies et al., 2002.

the DS and the RR-PMCs was inhibited by K02288 treatment. Many of the 38 PMC-enriched mRNAs that were regulated in a positive manner by BMP signaling encode proteins that regulate biomineralization, including spicule matrix proteins (sm29, sm30a, sm30f and Clect), secreted matrix metalloproteases (mmp16 and mmp24), p58b, and MSP130 family proteins (msp130rel1 and msp130rel3). Another BMP-regulated gene, otop2L, is a PMC-specific proton transporter that de-acidifies the cytoplasm of PMCs by removing protons that are liberated during the formation of $CaCO_3$ and is necessary for skeletogenesis (Chang et al., 2021). otop2L mRNA is concentrated in the dorsal region, although the protein is more widely distributed within the PMC syncytium (Sun and Ettensohn, 2014; Chang et al., 2021). These examples strongly suggest that BMP signaling supports dorsal skeletal growth through its positive regulation of a subset of biomineralization proteins. We suspect that other PMC-specific proteins that are sensitive to K0228

treatment, but which currently have unknown functions (see Tables 1, 2), also regulate biomineralization.

To identify candidate regulators of BMP signaling in the DS, we carried out a comprehensive analysis of the spatiotemporal expression patterns of the BMP ligands and receptors present in sea urchins, with a special focus on the expression of these genes during Phase 2 (Fig. 4, Fig. S1). We found that alk1/2 was strongly expressed by all PMCs during early development, but later became restricted within the syncytium to sites of active skeletal growth. Based on these findings, we hypothesize that Alk1/2 is the principal Type I BMP receptor that mediates BMP signaling responses within the PMC syncytium. The single Type II BMP receptor, bmpr2, showed a much less dynamic pattern of expression. We hypothesize that this receptor is present throughout the syncytium and that local differences in alk1/2 expression modulate local responses. In addition, we have confirmed the previous observation that bmp2/4 is

**Table 2. PMC-enriched mRNAs downregulated by axitinib+K02288**

| Echinobase gene symbol | Lv LOC ID | Function |
|---|---|---|
| Uncharacterized | LOC121415626 | Acanthoscurrin-like, antimicrobial peptide (predicted) |
| p58b[1] | LOC121425019 | Biomineralization, Type I transmembrane |
| LOC764585 | LOC121422431 | Heparan sulfate glucosamine 3-O-sulfotransferase 1 (predicted) |
| clec19a[2] | LOC121427330 | Lectin C-type domain, echinoidin-like |
| LOC579383[2] (msp130rel3) | LOC121427467 | Mesenchyme-specific cell surface glycoprotein |
| LOC373517[2] (msp130rel1) | LOC121427374 | Mesenchyme-specific cell surface glycoprotein |
| LOC115921620 & LOC105441121 | LOC121427517 | Mucin-like protein, glycoprotein (predicted) |
| LOC591784 | LOC121423895 | Phospholipase A2 (predicted) |
| LOC582939 | LOC121432263 | Rho GTPase activating protein (predicted) |
| sm30a[3] | LOC121406020 | Spicule matrix protein, biomineralization |
| LOC581461 (clect) | LOC121406083 | Spicule matrix protein, biomineralization, Lectin C-type domain (predicted) |
| LOC587674 | LOC121410904 | Sulfate Transporter (predicted), prestin-like |
| hhex | LOC121424894 | Transcription Factor, Homeodomain (predicted) |
| LOC100891544 | LOC121410060 | Translation Initiation Factor IF-2 (predicted) |
| prom1 | LOC121424710 | Transmembrane glycoprotein, localized to cilia/microvilli, differentiation, mutated in cancer cells (predicted) |
| LOC115919257 | LOC121411544 | Unknown |
| LOC100889313 | LOC121432130 | Unknown |
| LOC577899 | LOC121407396 | Unknown |
| LOC586717 | LOC121431115 | Unknown, aligns to unplaced and unannotated scaffold |
| uncharacterized | LOC121413488 | Unknown, does not align well to any S. purpuratus genes, annotated as ncRNA on NCBI |
| LOC100894023 | LOC121410563 | Unknown, Immunoglobulin domain, Fibronectin type III domain |

[1]Adomako-Ankomah and Ettensohn, 2011; [2]Illies et al., 2002; [3]Wilt et al., 2013.

**Table 3. PMC-enriched mRNAs downregulated specifically by axitinib**

| Echinobase gene symbol | Lv LOC ID | Function |
|---|---|---|
| angpt1l | LOC121428545 | Angiogenesis, angiopoietin 1-like (predicted) |
| LOC764069 | LOC121409286 | ATPase, phospholipid transporting ATPase (predicted) |
| SpP19L[1] | LOC121428634 | Biomineralization |
| P16[1,2] | LOC121416200 | Biomineralization, transmembrane |
| p58a[3] | LOC121425028 | Biomineralization, Type I transmembrane |
| LOC762549 | LOC121410970 | Gamma-aminobutyric acid receptor subunit gamma-2-like (Predicted) |
| LOC576560 | LOC121431448 | LamG domain, EGF domain, cell adhesion |
| LOC586753 | LOC121409093 | Matrix metalloproteinase 17, biomineralization, transmembrane |
| LOC586737 | LOC121409094 | Matrix metalloproteinase |
| LOC115919351[4] | LOC121410434 | Metalloproteinase inhibitor 3-like, extracellular matrix formation |
| LOC115918623 | LOC121425008 | Mucin-5A-like (predicted) |
| fgfr2l[5] | LOC121420312 | Myoblast growth factor receptor egl-15-like, biomineralization, transmembrane |
| LOC590168 | LOC121409284 | Myogenesis, smooth muscle, myb-like protein, smoothelin-like |
| LOC756397 | LOC121432068 | Neurofilament heavy polypeptide-like (predicted) |
| LOC756768 | LOC121424209 | Neurogenesis, neurotrypsin 2-like (predicted) |
| hck | LOC121430645 | Src family tyrosine kinase (predicted) |
| ets1 | LOC121419770 | Transcription Factor, ETS domain, biomineralization, EMT regulation |
| prox1 (prospero) | LOC121418266 | Transcription Factor, homeobox domain |
| myoD1l[6] | LOC121405904 | Transcription Factor, myogenic basic helix-loop-helix domain, muscle specification, ventral skeletal expression |
| tcf21l[7] (myoR2) | LOC121430479 | Transcription Factor, myogenic basic helix-loop-helix domain, ventral skeletal expression, blastocoelar expression |
| zcchc24 | LOC121431737 | Transcription Factor, zinc-finger domain, ETS domain (predicted) |
| LOC577685 | LOC121406053 | Translation Initiation Factor 2-like (predicted) |
| LOC594470 | LOC121431017 | Transmembrane protein (predicted) |
| LOC115919110 | LOC121411681 | Uncharacterized |
| LOC100893802 | LOC121405811 | Uncharacterized |
| LOC100891313 | LOC121416090 | Uncharacterized, glycine-rich |
| flt1[8,9] (vegfr-Ig10) | LOC121408625 | VEGF receptor, dorsal/ventral axis specification, biomineralization |

[1]Illies et al., 2002; [2]Cheers and Ettensohn, 2005; [3]Adomako-Ankomah and Ettensohn, 2011; [4]Clouse et al., 2015; [5]Röttinger et al., 2008; [6]Beach et al., 1999; [7]Andrikou et al., 2013; [8]Duloquin et al., 2007; [9]Sun and Ettensohn, 2017.

initially expressed by the ventral ectoderm during Phase 1 whereas *bmp5/8* is expressed more broadly, and both become strongly expressed in the DS during Phase 2 (Luo and Su, 2012). Thus, both ligands could contribute to the regulation of Phase 2 skeletogenesis on the dorsal side of the embryo.

Our findings show that BMP signaling plays an important role in regulating gene expression in the DS. However, because signaling is active in multiple dorsal tissues, and because K02288 blocks signaling in all tissues, there are several possible scenarios by which such regulation could occur. Importantly, the presence of high levels of pSMAD1/5/8 in dorsal PMCs shows that signaling is active in these cells (our findings and Lapraz et al., 2009; Chen et al., 2011; Haillot et al., 2015). One simple (and therefore attractive) hypothesis is that BMPs produced by PMCs function in an autocrine fashion to regulate gene expression in these cells, either by direct activation of skeletogenic effector genes by pSMAD1/5/8, or indirectly via SMAD-dependent expression of other TFs (see Fig. 8). Residual BMP2/4 produced by the ventral ectoderm but active in the DS might also contribute to pSMAD1/5/8 activation in the PMCs. Although we currently favor a direct, cell-autonomous role for pSMAD1/5/8 activation in controlling gene expression and skeletal growth in PMCs, primarily due to the simplicity of such a mechanism, we cannot exclude more complex scenarios. For example, BMP signaling and pSMAD1/5/8 might function in the dorsal ectoderm to activate other signals (different from BMPs) that act on PMCs to control gene expression and skeletal growth, either alone or in combination with BMPs. Other, even more complex, roles for BMP signaling can be envisioned. Additionally, the relative contributions of the two BMP ligands, BMP2/4 and BMP5/8, to dorsal signaling have yet to be teased apart. Clearly, additional studies will be required to dissect the mechanisms by which BMP signaling regulates gene expression and skeletal growth in dorsal PMCs.

K02288 inhibits the expression of *bmp2/4* and *bmp5/8* in the DS during Phase 2 (Fig. 5G-H′,J-K′). This finding supports previous evidence that *bmp2/4* expression is autoregulated in a positive fashion by the BMP pathway in the syncytium (Luo and Su, 2012) and indicates that *bmp5/8* is similarly regulated. The shift in localized ligand expression also coincides with the transition from Phase 1 skeletogenesis to Phase 2. The initial activation of BMP ligand expression in dorsal PMCs could be driven by BMPs that are produced by the ventral ectoderm during early development and active dorsally, or by localized cues other than BMPs. As discussed above, the increase in expression of BMP ligands in dorsal PMCs might be driven by BMPs secreted by the PMCs themselves via an autocrine, positive regulatory mechanism. Notably, in addition to the two BMP ligands, our RNA-seq analysis identified *smadIP* as a gene upregulated by BMP signaling, and previous work showed that *smadIP* is expressed selectively by dorsal PMCs (Valencia, 2018). Thus, *smadIP* is an additional component of the BMP signaling pathway regulated in a positive manner by BMP signaling. Studies in other organisms have shown that the regulation and function of *smadIP* are complex (Hegarty et al., 2015). SmadIP can act as a repressor or an activator of expression depending on co-effector binding and can interact directly with receptor-activated Smad proteins. Further studies will be required to elucidate the function of *smadIP* in the DS.

Notably, axitinib treatment specifically downregulated *vegfr-Ig10* expression (Table 3) and the expression intensity of *vegfr-Ig10* was reduced when visualized by HCR (Figs 5, 6). This suggests that, similar to BMP signaling, continuous positive VEGF signaling regulates the VEGF signaling pathway itself during skeletogenesis.

Our work has also shown that the distal-tip RR-PMCs of *L. variegatus* share many features with the DS. The RRs were the only other skeletal structure to show a striking morphological defect

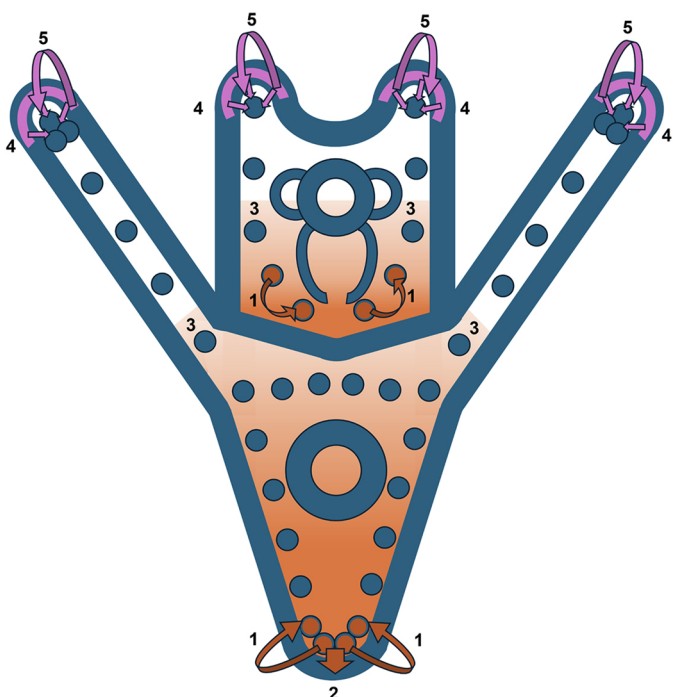

**Fig. 8. Provisional model of signaling pathway contributions to dorsal/ ventral skeletal growth.** BMP ligands (orange) are expressed by the RR-PMCs and dorsal (scheitel) PMCs. Autocrine BMP signaling acts to maintain expression of BMP ligands in these cells (1). Scheitel PMCs export BMP ligands to the overlying dorsal ectoderm, activating SMAD1/5/8 in this tissue (2). BMP signaling may weakly contribute to ventral skeletal growth via a gradient of BMP ligands originating from the dorsal skeletal elements that signals to *alk1/2*-expressing cells at the growing tips of the arms (3). VEGF3 ligand is exported from the ectoderm overlying the arm tips and VEGF signaling (magenta) is essential for elongation of the ventral skeletal elements but not the dorsal elements (4). VEGF signaling maintains *vegfr-Ig10* expression in the PMCs at the tips of the post-oral and anterolateral rods (5).

in response to K02288, as treatment with the drug prevented the distal branching event in the RRs (Fig. 3). Additionally, several mRNAs with elevated expression in the dorsal PMCs, including *gata3*, *scl* and *sm29*, were also enriched in RR PMCs. During Phase 2, the RRs, like the DS, become signaling centers that express high levels of both BMP ligands. The production of BMP ligands by PMCs in the RRs, which are located closer to the ventral side of the embryo than the dorsal PMC chain, could explain the limited role that BMP signaling plays in ventral skeletal development in *L. variegatus*. It should be noted that RRs do not form in all sea urchin species. Although the directionality of the evolutionary change is unknown in this case, one hypothesis is that a spatial redeployment of the dorsal PMC GRN occurred at the distal end of the dorsal-ventral connecting rod, leading to the formation of the RR. Species without RRs may experience less pronounced effects on ventral skeletal development when treated with K02288 than we observed in *L. variegatus*.

The ligands that activate BMP and VEGF signaling are located on opposite sides of the embryo during Phase 2, and to a large extent these two pathways operate independently of one another in controlling gene expression and skeletal growth (Fig. 8). The skeletal defects observed following disruption of these two pathways during Phase 2 are very different from one another (Adomako-Ankomah and Ettensohn, 2013; this study). In addition, approximately two-thirds (49/74) of the PMC dGRN genes that we found to be signal

dependent during Phase 2 were sensitive to either K02288 or axitinib, but not to both inhibitors (Fig. 7). A previous study also identified ventral-specific effects of axitinib on the expression of several genes expressed at sites of active skeletal growth (Sun and Ettensohn, 2014). Notably, we identified several TF-encoding genes that were regulated in a pathway-specific manner during Phase 2. We found that *gata3* and *scl*, two genes expressed selectively in dorsal PMCs (Solek et al., 2013), were regulated in a positive manner by BMP signaling, while *myoD* and *myoR2*, which are expressed selectively in ventral PMCs (Beach et al., 1999; Valencia, 2018), were upregulated by VEGF signaling (Tables 2, 3). These signal-dependent TFs may contribute to the region-specific expression of downstream effector genes in the dorsal and ventral regions. A major unanswered question is whether the distinctive suites of BMP-dependent and VEGF-dependent biomineralization genes expressed in the dorsal and ventral regions of the PMC syncytium contribute to the distinct morphologies of the skeletal elements that develop in these regions, i.e. the linear skeletal rods that support the ventral arms and the branched scheitel that forms dorsally.

We also identified genes (25/74) that were regulated by both BMP and VEGF signaling. There are examples of genes that are expressed at high levels in PMCs both at the tips of the post-oral arms and the dorsal chain (Sun and Ettensohn, 2014), suggesting that some genes may be regulated in a positive manner by both signaling pathways. The expression pattern of *gata3*, however, reveals another possible mechanism. In untreated embryos, *gata3* expression was restricted to PMCs in the dorsal chain, as previously reported (Solek et al., 2013). In axitinib-treated embryos, however, *gata3* mRNA was observed throughout the PMC syncytium. Because mRNAs typically have very limited ability to diffuse through the PMC syncytium (see Khor et al., 2023), we presume that the expanded domain of *gata3* mRNA in axtinib-treated embryos reflects an expanded region of *gata3* transcription. These findings suggest that, although BMP signaling is required for the dorsal expression of *gata3* (see Figs 5, 6), VEGF-mediated repression in the ventral region also plays an important role in restricting *gata3* expression to the dorsal region. The mechanism of this repression, and the activators that drive *gata3* expression on the ventral side of axitinib-treated embryos, are currently unknown.

Our RNA-seq analysis showed that only about 20% of late, PMC-enriched mRNAs were significantly affected by treatment with axitinib or K02288. Thus, the striking, localized effects of these pathways on skeletal growth are mediated through their control of relatively limited sub-circuitry within the PMC dGRN. Conversely, ~80% of late, PMC-enriched mRNAs were not significantly affected by treatment with axitinib or K02288, a finding consistent with our HCR *in-situ* hybridization analysis, which showed that genes such as *sm37* and *sm29* were still expressed throughout the syncytium in the presence of either drug. These findings suggest the presence of non-localized signals that regulate gene expression throughout the PMC syncytium or cell-autonomous mechanisms that drive much of the skeletogenic network independently of localized signals. Other signaling mechanisms are known to regulate skeletogenesis and may contribute to the regulation of the PMC dGRN during Phase 2. FAK-ROCK-ERK signaling allows for mechanosensing and regulates skeletal growth and branching (Hijaze et al., 2024; Layous et al., 2025). Ionic gradients are also able to modulate responses to signaling pathways through the activity of voltage-gated sodium channels allowing for general regulation of skeletal growth (Thomas et al., 2023). TGFβ signaling also contributes to skeletogenesis, especially the formation of anterior skeletal elements (Piacentino et al., 2015; Sun and Ettensohn, 2017). The activity of these signaling pathways

does not appear to be polarized along the dorsoventral axis during Phase 2, however, and disruption of these pathways does not generate dorsal- and ventral-selective defects similar to those observed after disruption of VEGF or BMP signaling. Therefore, we propose that VEGF and BMP signaling are the key signaling pathways that regulate skeletal growth during post-gastrula embryogenesis on the ventral and dorsal sides of the skeleton, respectively.

## MATERIALS AND METHODS

### Animals
Adult *S. purpuratus* were acquired from Peter Halmay (San Diego Fishermen's Working Group, 11103 Highway 67, Lakeside, CA 92040-1407, USA). Adult *L. variegatus* were acquired from either Duke University (NC, USA) or from Pelagic (Sugarloaf Key, FL, USA). Spawning was induced by intracoelomic injection of 0.5 M KCl. Embryos were cultured in ASW at 15°C (*S. purpuratus*) or 18-23°C (*L. variegatus*) in temperature-controlled incubators.

### Drug treatments
A 5 mM stock solution of axitinib (Selleckchem) was prepared in DMSO and stored at −20°C. *L. variegatus* were cultured in a final concentration of 75 nM axitinib in ASW.

A 14.2 mM stock solution of K02288 (Sigma-Aldrich, SML1307-5MG) was prepared in DMSO and stored at 4°C. Both *S. purpuratus* and *L. variegatus* embryos were cultured in a final concentration of 1 μM K02288 in ASW. To observe early developmental effects, eggs from both species were fertilized in 0.5 μM K02288 ASW and cultured continuously in the presence of the drug. To observe late developmental effects, embryos from both species were cultured in untreated ASW before being transferred into 1 μM K02288 ASW at later stages.

For RNA-seq and HCR experiments, *L. variegatus* were initially cultured in untreated ASW and then transferred into of 75 nM axitinib in ASW or 1 μM K02288 in ASW after the initial tri-radiate spicule rudiments had formed (~18-20 hpf when cultured at 20-21°C). Embryos were incubated continuously in drug-treated ASW until the late pluteus stage (2 dpf) when they were either fixed for HCR or lysed for total RNA collection. To observe the effect of K02288 treatment on pSmad1/5/8 activation, *S. purpuratus* embryos were initially cultured in untreated ASW and then transferred into 1 μM K02288 ASW after the PMC syncytium had formed and the embryos were beginning to gastrulate (28 hpf when cultured at 15°C). *S. purpuratus* embryos were fixed and immunostained at the late prism/early pluteus stage (2 dpf).

Control embryos were incubated in ASW containing a volume of DMSO equal to the highest volume used and treated at the same developmental stages for any given replicate set of drug-treated embryos.

### Fluorescence *in situ* HCR
Embryos were fixed for 1 h in 4% paraformaldehyde in ASW and stored in 100% methanol at −20°C. HCR probes and fluorescent hairpins were purchased from Molecular Instruments®. Fixed embryos were rehydrated with three washes of RNase-free water mixed with progressively lower concentrations of methanol (70%, 50%, 30%) and stained following the Molecular Instruments published protocol for sea urchins (Choi et al., 2016).

### Imaging
Images were collected using an Olympus BX60 microscope fitted with a 20× dry objective (N.A., 0.7), an X-Cite XYLIS LED light source (Excelitas Technologies) and a Xyla 4.2 sCMOS camera (Oxford Instruments). Images were processed using cellSens imaging software (Olympus) and Fiji/ImageJ version 2.16.0/1.54p (Schindelin et al., 2012).

### RNA-seq and bioinformatics analysis
Three replicate sets of embryos resulting from separate matings were used for RNA-seq. Each replicate set contained three samples consisting of embryos treated with DMSO, 75 nM axitinib or 1 μM K02288 starting after formation of tri-radiate spicule rudiments as described above. Late stage plutei (2 dpf, 21°C) were then collected by centrifugation (160 *g* for 30 s) and excess ASW was

removed. Embryos were lysed and total RNA was extracted using a Pure Link™ RNA Mini Kit (Invitrogen) following the manufacturer's protocol. Samples were then treated with Turbo DNase™ (Invitrogen) and LiCl precipitated to remove genomic DNA. RNA samples were then shipped to Novogene Corporation Inc. for sequencing. mRNA was purified by Novogene using poly-A enrichment purification, strand-specific cDNA synthesis was performed, and sequenced by synthesis using the Illumina sequencing platform.

Raw reads were mapped to the *L. variegatus* genome (v3.0) using the HISAT2 alignment program (Mortazavi et al., 2008). Gene expression levels were then quantified as fragments per kilobase of transcript sequence per millions base pairs sequenced (FPKM). Differentially expressed genes between treatment groups were identified using DEseq2 software (Love et al., 2014). A DESeq2 $P \leq 0.05$ with no minimum log2FoldChange cutoff (e.g. |log2FoldChange|≥0.0) were used as threshold cutoffs to determine differentially expressed genes. clusterProfiler software (Yu et al., 2012) was used for gene ontology analysis of differentially expressed genes.

To generate a set of PMC-enriched genes, we re-clustered scRNA-seq data from Massri et al. (2021) from four time points (16, 18, 20 and 24 hpf), representing the late gastrula-pluteus stages, and identified the PMC cluster based on a large set of known, PMC-specific mRNAs (Rafiq et al., 2014). We focused on genes that were expressed in at least 25% of the cells in the PMC cluster and that exhibited a log2-fold enrichment of >2 relative to non-PMC cells. Using these criteria, we identified a collection of 321 genes that were differentially expressed by PMCs during Phase 2. We supplemented this gene set with an additional 46 genes that were shown in previous studies to be expressed selectively by PMCs during late embryogenesis, including a set of 29 TF-encoding genes (Solek et al., 2013; Sun and Ettensohn, 2014; Valencia, 2018; Massri et al., 2021). This set of genes was then compared to the list of genes that were differentially expressed in response to either axitinib or K02288 treatment in our bulk RNA-seq dataset.

### Skeletal element measurements
*L. variegatus* embryos treated with DMSO or 1 μM K02288 starting after tri-radiate formation as described above were live imaged at the late pluteus stage (2 dpf). Using a combination of DIC and polarized light images across multiple focal planes the anterolateral rods, RRs, ventral-transverse rods, post-oral rods and body rods of each embryo was imaged. The lengths of these five skeletal elements were measured on both sides of the embryo using the segmented line tool of Fiji/ImageJ. Anterolateral rods and RRs were measured from their distal tips to the junction with the dorsal-ventral connecting rod. Ventral-transverse rods were measured from their tips to the center of the branch point where the post-oral rods and body rods begin. Post-oral rods were measured from their tips to the center of the same branch point. Body rods were measured from the center of the same branch point to the branch point where the skeleton bends to form the scheitel. Embryos resulting from two separate matings were imaged and measured for this comparison.

### Sp-pSmad1/5/8 immunostaining
Phospho-SMAD (Ser463/465) (41D10) Rabbit mAb (Cell Signaling Technology, 9516) was the generous gift of T. Lepage (Institut de Biologie Valrose, University of Nice, Nice, France). *S. purpuratus* embryos were fixed at various developmental stages for 15 min in 4% paraformaldehyde in ASW at room temperature and then immediately immunostained according to the protocol described by Haillot et al. (2015). Of note, this antibody is unable to detect pSmad1/5/8 in *L. variegatus*, likely because the amino acid sequence near the phosphorylation site is slightly different than the corresponding site in *S. purpuratus*. As such, pSmad1/5/8 enrichment and the effects of K02288 on the phosphorylation of Smad1/5/8 were only analyzed in *S. purpuratus*.

### Statistical analysis
To determine the effects of K02288 on the growth of various skeletal elements the average length of each element was determined for DMSO-treated and 1 μM K02288-treated embryos. The average lengths were then compared using a two-tailed Student's *t*-test with unequal variance in Excel. A *P*-value of <0.05 was used as a cutoff for significance.

For all statistical analyses, asterisks represents the following *P*-values: *$P$<0.05, **$P$<0.01, ***$P$<0.005, ****$P$<0.001, *****$P$<0.0005.

## Acknowledgements
We thank fellow lab members Xiantong Xin and Xinyi Li for their assistance in re-clustering the scRNA-seq data from Massri et al. (2021) and matching genes from this dataset to the current *L. variegatus* genome (v3.0). We are also very grateful to Dr Thierry Lepage (Institut de Biologie Valrose) for providing anti-pSmad1/5/8 antibody and an immunostaining protocol for this antibody. Finally, we thank Dr Dave McClay (Duke University) for his assistance in obtaining adult sea urchins, including showing one of us (W.B.D.) how to trawl.

## Competing interests
The authors declare no competing or financial interests.

## Author contributions
Conceptualization: C.A.E., W.B.D.; Data curation: W.B.D.; Formal analysis: W.B.D.; Funding acquisition: C.A.E., W.B.D.; Investigation: W.B.D.; Methodology: W.B.D.; Project administration: C.A.E.; Supervision: C.A.E.; Validation: W.B.D.; Visualization: W.B.D.; Writing – original draft: W.B.D.; Writing – review & editing: C.A.E., W.B.D.

## Funding
This work was supported by the National Science Foundation (IOS2004952 to C.A.E.) and by the National Institutes of Health (1090910-F31GM14918502-NIGMS to W.B.D.). Open Access funding provided by Carnegie Mellon University. Deposited in PMC for immediate release.

## Data and resource availability
Raw read and processed files for bulk RNA-seq are available in Gene Expression Omnibus under accession number GSE324596. All other relevant data and details of resources can be found within the article and its supplementary information.

## Peer review history
The peer review history is available online at https://journals.biologists.com/dev/lookup/doi/10.1242/dev.205344.reviewer-comments.pdf

## Special Issue
This article is part of the Special Issue 'The Extracellular Environment in Development, Regeneration and Stem Cells', edited by Alex Hughes and Rashmi Priya. See related articles at https://journals.biologists.com/dev/issue/153/16.

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
