## [Peer Review File · Development (Cambridge, England)]

BMP signaling regulates dorsal skeletal growth in the sea urchin embryo

William B. Douglas and Charles A. Etensohn

DOI: 10.1242/dev.205344

Editor: Swathi Arur

Review timeline

Original submission:	3 November 2025
Editorial decision:	5 January 2026
First revision received:	21 January 2026
Editorial decision:	9 February 2026
Second revision received:	18 February 2026
Accepted:	23 February 2026

Original submission

First decision letter

MS ID#: dev.205344

MS TITLE: BMP signaling regulates dorsal skeletal growth in the sea urchin embryo

AUTHORS: William Douglas and Charles A. Etensohn

Dear Dr Etensohn,

I have now received all the referees' reports on the above manuscript, and have reached a decision. The referees' comments are appended below.

As you will see, the referees express considerable interest in your work, and provide recommendations to further improve the manuscript. If you are able to revise the manuscript along the lines suggested, which may involve further experiments, I will be happy receive a revised version of the manuscript. Your revised paper will be re-reviewed by one or more of the original referees, and acceptance of your manuscript will depend on your addressing satisfactorily the reviewers' major concerns.

Please attend to all of the reviewers' comments and ensure that you upload both a 'clean' version of your Word file, along with a highlighted version clearly showing where you have made changes in the revised manuscript. Please avoid using 'Tracked changes' in Word files as these are lost in PDF conversion. I should be grateful if you would also provide a point-by-point response detailing how you have dealt with the points raised by the reviewers in the 'Response to Reviewers' box. If you do not agree with any of their criticisms or suggestions please explain clearly why this is so.

Reviewer 1

Advance summary and potential significance to field

Here a BMP inhibitor, K02288, was used to perturb the BMP pathway and show a number of effects on skeletogenesis, most strikingly in the dorsal skeleton. The authors examined BMP receptors and

ligands for those most affected by the drug inhibition. They show that the body rods are the most sensitive to BMP inhibition and also show effects on other rods, though to a lesser degree. They also examined some of the transcription factors thought to be part of the GRN of the skeletogenic cells. Some of these also were affected. Further, they generated RNA-seq libraries of inhibited embryos and compared the outcome with a published scRNA-seq library where skeletogenic cells were readily identified, and a previous analysis for additional genes. They noted that many of the genes expressed in late skeletogenic cells are affected by the BMP inhibition indicating that BMP, especially BMP2/4, is a major signaling contributor to dorsal skeletogenesis.

Comments for the author

This manuscript reports on BMP signaling during skeletogenesis in the sea urchin larva. Previous research on signaling focused on VEGF from the ectoderm that is an important signal that activates skeletal production by the primary skeletogenic cells (PMCs). Little effort previously was spent to understand how BMP signaling might also contribute to skeletogenesis. Here a BMP inhibitor, K02288, was used to perturb the BMP pathway and show a number of effects on skeletogenesis, most strikingly in the dorsal skeleton. The authors examined BMP receptors and ligands for those most affected by the drug inhibition. They show that the body rods are the most sensitive to BMP inhibition and also show effects on other rods, though to a lesser degree. They also examined some of the transcription factors thought to be part of the GRN of the skeletogenic cells. Some of these also were affected. Further, they generated RNA-seq libraries of inhibited embryos and compared the outcome with a published scRNA-seq library where skeletogenic cells were readily identified, and a previous analysis for additional genes. They noted that many of the genes expressed in late skeletogenic cells are affected by the BMP inhibition indicating that BMP, especially BMP2/4, is a major signaling contributor to dorsal skeletogenesis.

The manuscript is well written though there are issues with the figures and/or figure legends that must be improved. These are indicated below.

1. You don't seem to include bars to indicate length in the images. Since the figures show embryos at different magnification and in some cases partial embryos this is helpful. Also, since you use Lv and Sp the measurement bars will help since the size of these embryos is different.
2. In Figure 3 - don't you mean μm instead of μM which is micromolar?
3. I can understand in Fig. 3 that the K02288 selectively affects the body rods but the recurrent rods don't appear to be any more inhibited than the other rods you suggest are less affected by the K02288.
4. The Fig. 4 puzzles me. You state that the staining shows autofluorescence but if that were the case all the embryos would show the autofluorescence since the only difference in the protocol you use is the mRNA targeted. However the BMP2/4 doesn't show any. And most BMP5/8 don't show any so you might want to reconsider that cause. Also, it is hard to tell the orientation of the embryos in this figure. It would be most helpful to indicate orientation. Also, L,L' and P,P' are the same embryo and only one of the images seems to have what you call autofluorescence. Also M,M' and Q,Q' are the same embryo. Apparently you pseudocolored two different HCR probes. It would be helpful to indicate when double stained embryos are indicated.
5. Figure 5. It would help if the rows are separated by a white line. It took a while for me to figure out that the top row is all SM37 Gata3 since the perturbed pairs aren't labeled. Also, are the embryos in control vs perturbed states shown in the same orientation? For interpretations to be most accurate they should be because if you look at one embryo where the dorsal surface is up and another in which the ventral surface is up, the relative stain of indentially treated embryos often is different because of the depth of field. In the images shown I can't attest to that equality but a DV or a VV would be helpful to confirm. This is also crucial in your calls since it is clear that fluorescence levels in images is semiquantitative at best, so identical orientation is often crucial. I don't understand C'. You say that the orange arrowhead (looks yellow to me) shows ectopic expression, yet the control seems to show expression in the same area.
6. Fig. 5 is labeled post-triradiate late pluteus while Fig. 6 embryos are labeled triradiate late pluteus embryos. What is the difference? The triradiate stage you show in Fig.1 is at late gastrula or early prism at the latest. Also, in both Fig.5 and 6 check the alignment letters in your Figure legends I think you indicate the wrong letters in both figure legends. And surely there is a mistake since there is no R image.

Reviewer 2*Advance summary and potential significance to field*

The purpose of this study is to identify the external signal that governs dorsal skeleton formation during the second phase of skeletogenesis in the sea urchin embryo. Prior work from the senior author's lab showed that VEGF3 is important for skeletal formation broadly, and more specifically in the vegetal half of the embryo. The present study builds on these results, focusing on the role of BMP in regulating dorsal skeletogenesis. BMP was considered a candidate due to its well-defined function in dorsal-ventral specification during embryogenesis. To investigate the role of BMP in skeletogenesis, the authors used the small molecule inhibitor K02288 to inhibit the Type 1 BMP receptors required for transduction of BMP signals. Quantification of skeletal elements after K02288 treatment inhibition shows a significant decrease in multiple skeletal elements, excluding ventral-transverse rods and preferentially affecting dorsal elements. Analysis of downstream BMP-responsive genes, including transcription factors (*gataC*, *scl*), BMP ligands (*bmp2/4*), and PMC biomineralization genes, reveals that K02288 treatment frequently decreased expression specifically within the dorsal PMC syncytium. Finally, the authors conduct bulk RNA sequencing on embryos treated individually with K02288 or axitinib (a VEGF inhibitor) and DMSO-treated controls across two stages of skeletogenesis to identify unique and overlapping contributions to skeletogenesis from BMP and VEGF pathways. This experiment identified 74 genes enriched in PMCs during phase 2 skeletogenesis affected by drug treatment, with 25 of these affected by both treatments, suggesting some overlap between these pathways.

While the development of the embryonic skeleton of sea urchins has been studied fairly extensively, this manuscript provides the first substantive investigation into molecular mechanisms that pattern the more complex larval skeleton.

Comments for the author

The experiments are well designed and executed. While it would have been nice to have data from a more targeted knock-down approach, the authors have taken care to validate the specificity of K02288 treatments. Experimental results are well documented, and support the main conclusions. Overall, there no serious concerns.

Minor issues that should be addressed before publication:

- Each figure with an embryo should include at least one scale bar.
- It may be helpful to provide cues about body axis orientation within the figures as opposed to solely in the caption to help orient readers unfamiliar with sea urchin embryos.
- Figure 2A/A' and B/B' show a direct comparison of a control and treated embryo, but they are in different orientations. It may again be helpful to explicitly show this difference in orientation within the body of the figure or provide images with the same position for a clearer comparison of skeletal structure.
- Figure 2 provides results to support the use of K02288 as an alternative to morpholino usage. However, no direct comparison of knockdowns is shown to support the efficacy and specificity of this method. The authors would benefit from providing images of these comparisons so readers can see the difference in penetrance between these methods.

First revisionAuthor response to reviewers' comments

We thank the two reviewers for their evaluations of the manuscript and valuable recommendations. We have made changes in response to each of their comments and feel that the revised manuscript has been significantly improved. All changes to the manuscript in response to the reviewers' suggestions have been highlighted in **blue font**.

Below are point-by-point responses to the concerns raised by the reviewers.

Reviewer 1:

The manuscript is well written though there are issues with the figures and/or figure legends that must be improved. These are indicated below.

1. You don't seem to include bars to indicate length in the images. Since the figures show embryos at different magnification and in some cases partial embryos this is helpful. Also, since you use Lv and Sp the measurement bars will help since the size of these embryos is different.

We thank the reviewer for noting this. Scale bars have been added to all figures containing images of embryos.

2. In Figure 3 - don't you mean μm instead of μM which is micromolar?

We thank the reviewer for noting this- it has been corrected.

3. I can understand in Fig. 3 that the K02288 selectively affects the body rods but the recurrent rods don't appear to be any more inhibited than the other rods you suggest are less affected by the K02288.

We have revised the figure legend and Results section to clarify this point. As seen in Figure 3B, the two recurrent rods in *L. variegatus* project dorsally from the anterior ends of the dorsoventral connecting rods. After a period of linear growth, each recurrent rod begins to grow at an angle relative to the initial (proximal) segment. The proximal and distal segments, with the transition between them marked by a sharp bend, make up the recurrent rod. In embryos treated with K02288, the overall length of the recurrent rods is reduced, but an even more striking change is that they are straight rather than angled. These effects on the growth of the recurrent rods were notable and led us to the conclusion that BMP signaling plays an important role in patterning this rod.

4. The Fig. 4 puzzles me. You state that the staining shows autofluorescence but if that were the case all the embryos would show the autofluorescence since the only difference in the protocol you use is the mRNA targeted. However the BMP2/4 doesn't show any. And most BMP5/8 don't show any so you might want to reconsider that cause. Also, it is hard to tell the orientation of the embryos in this figure. It would be most helpful to indicate orientation. Also, L,L' and P,P' are the same embryo and only one of the images seems to have what you call autofluorescence. Also M,M' and Q,Q' are the same embryo. Apparently you pseudocolored two different HCR probes. It would be helpful to indicate when double stained embryos are indicated.

We're fortunate that when using HCR-FISH, genuine signal is easily distinguished from background. The former is distinctly punctate, while background signal has a foggy, relatively weak, due to relatively low expression of the target mRNA or trial-to-trial variability in the staining intensity) or because the orientation of the embryo required us to image deeply in the tissue. The wavelengths used also affected the amount of background signal to some extent. Thus, there are various reasons why the background signal in our images varies in intensity, but when scoring embryos, background can easily be distinguished from genuine signal.

In response to the reviewer's other concerns, labels have been added to denote the orientation of embryos, and images that show different fluorescent channels (and thus different mRNAs) in the same embryo labeled with multiple probes have been noted in the figure legend.

5. Figure 5. It would help if the rows are separated by a white line. It took a while for me to figure out that the top row is all SM37 Gata3 since the perturbed pairs aren't labeled. Also, are the embryos in control vs perturbed states shown in the same orientation? For interpretations to be most accurate they should be because if you look at one embryo where the dorsal surface is up and another in which the ventral surface is up, the relative

stain of identically-treated embryos often is different because of the depth of field. In the images shown I can't attest to that equality but a DV or a VV would be helpful to confirm. This is also crucial in your calls since it is clear that fluorescence levels in images is semiquantitative at best, so identical orientation is often crucial. I don't understand C'. You say that the orange arrowhead (looks yellow to me) shows ectopic expression, yet the control seems to show expression in the same area.

We thank the reviewer for this suggestion, and individual rows have been visually separated.

Images in the manuscript that show control and perturbed states were not chosen to match based on which surface was up towards the objective. When embryos were scored under the microscope, however, all focal planes were examined and our scoring included a mixture of embryos viewed from the dorsal and ventral sides. As the control staining is extremely strong, easily identifiable, and consistent (~90% for *gata3* A' which is the lowest control value), and we scored many embryos at multiple focal planes, we are confident that the orientation of the embryos did not affect our determination of the effect of K02288 on mRNA expression. Additionally, only embryos with clear PMC control staining (*sm37/vegfr-ig10*) were scored for any detectable expression of the mRNA of interest in the dorsal PMCs. Embryos were also scored for any expression (even faint) that matched the strong, control expression pattern, or for complete lack of detectable expression. Under these stringent scoring requirements, we found that most embryos showed dramatic changes in *gata3*, *scl*, *bmp2/4*, and *bmp5/8* expression.

The legend has been changed to indicate that the arrowhead in C' is yellow.

With regard to the ectopic expression, control embryos (A') show *gata3* expression in the coelomic pouches, a structure that fails to form in both K02288- and axitinib-treated embryos. In A' the expression from the coelomic pouches (above ventral gut expression) does not overlap with the PMC staining of *sm37*. A' also shows two PMCs slightly lower and to the left and right of the ventral gut which are the recurrent rod PMCs. In C', several PMCs (5 - 6) that form the anterolateral and dorsal-ventral connecting rod are strongly expressing *gata3* which is never seen in control embryos. Additionally, all of the PMCs of the body rod are expressing *gata3* strongly when it is typically only present in the lower half of the body rod (A'). The yellow arrowhead was intended to indicate that *gata3* expression had reached all the way to distal end of the anterolateral rod PMCs in axitinib-treated embryos. Additional arrowheads have been added to mark other PMCs that are ectopically expressing *gata3*.

6. Fig. 5 is labeled post-triradiate late pluteus while Fig. 6 embryos are labeled triradiate late pluteus embryos. What is the difference? The triradiate stage you show in Fig.1 is at late gastrula or early prism at the latest. Also, in both Fig.5 and 6 check the alignment letters in your Figure legends I think you indicate the wrong letters in both figure legends. And surely there is a mistake since there is no R image.

We thank the reviewer for detecting both of these errors. Figs. 5 and 6 have been adjusted to both state tri-radiate stage - late pluteus. As described in the Materials and Methods, drug treatment began after the tri-radiates had visibly formed. The letter alignments of the legends for Fig. 5 and 6 have also been corrected and now match the letters present on the image panels.

Reviewer 2:

Minor issues that should be addressed before publication:

- Each figure with an embryo should include at least one scale bar.
Scale bars have been added to all figures containing images of embryos.
- It may be helpful to provide cues about body axis orientation within the figures as opposed to solely in the caption to help orient readers unfamiliar with sea urchin embryos.
Dorsal/ventral axis labels have been added to figure images.
- Figure 2A/A' and B/B' show a direct comparison of a control and treated embryo, but they are in different orientations. It may again be helpful to explicitly show this

difference in orientation within the body of the figure or provide images with the same position for a clearer comparison of skeletal structure.

At the stage shown, the structure of the skeleton in control embryos makes it difficult to mount embryos such that they are oriented to provide a ventral view, while this view is much more common in treated embryos. A lateral view of a treated embryo has been added to provide a clearer frame of reference.

· Figure 2 provides results to support the use of K02288 as an alternative to morpholino usage. However, no direct comparison of knockdowns is shown to support the efficacy and specificity of this method. The authors would benefit from providing images of these comparisons so readers can see the difference in penetrance between these methods.

The effects of morpholino knockdowns to perturb BMP signaling in sea urchin embryos have been extensively documented in other, published studies (Luo and Su, 2012, Molina et al., 2013, Haillet et al., 2015). In particular, K02288 treatment after fertilization leads to morphological defects similar to those seen in Alk1/2 + Alk3/6 double knockdowns seen in Haillet et al. 2015. Based on the findings in those studies, we conclude that K02288 treatment is similarly effective and specific to the Type I BMP receptors: Alk1/2 and Alk3/6. Those studies are referenced in the body of the manuscript and readers can directly compare our results to the published morphant phenotypes.

References

- Haillet, E., Molina, M. D., Lapraz, F. & Lepage, T. 2015. The Maternal Maverick/GDF15-like TGF- β Ligand Panda Directs Dorsal-Ventral Axis Formation by Restricting Nodal Expression in the Sea Urchin Embryo. *PLoS Biol*, **13**, e1002247.
- Luo, Y. J. & Su, Y. H. 2012. Opposing nodal and BMP signals regulate left-right asymmetry in the sea urchin larva. *PLoS Biol*, **10**, e1001402.
- Molina, M. D., De Croz , N., Haillet, E. & Lepage, T. 2013. Nodal: master and commander of the dorsal-ventral and left-right axes in the sea urchin embryo. *Curr Opin Genet Dev*, **23**, 445-53.

Second decision letter

MS ID#: dev.205344R1

MS TITLE: BMP signaling regulates dorsal skeletal growth in the sea urchin embryo

AUTHORS: William Douglas and Charles A. Ettensohn

Dear Dr Ettensohn,

I have now received all the referees reports on the above manuscript, and have reached a decision. The referees' comments are appended below.

The overall evaluation is positive and we would like to publish a revised manuscript in Development, provided, the concerns pointed out by Reviewer 1 are addressed. In addition, I ask that the authors pay careful attention to grammar and citations of figures and references, since there seem to be several errors which will affect the rigor and clarity of the manuscript. While I do not expect to send the manuscript back to the reviewers, I will review it carefully, thus, please attend to all of the reviewers' comments in your revised manuscript and detail them in your point-by-point response. If you do not agree with any of their criticisms or suggestions explain clearly why this is so.

Reviewer 1*Advance summary and potential significance to field*

In reading this manuscript for the second time the story is clearer that BMP has a strong impact on the patterning of the dorsal skeleton, and also impacts the patterning of other skeletal rods. The unique feature of the role of BMP relative to previous publications is in determining how it signals and showing a group of downstream genes whose expression is impacted by the BMP signaling. For those reasons the manuscript has merit.

Comments for the author

In reading the revised manuscript, some of the concerns have been corrected, especially concerns over the Figure legends. Below, however there are still issues of concordance. In reading the text and paying close attention to the concordance with the Figures, there are a number of issues. They are given below. These need to be fixed so the text matches what is shown in the figures.

1. For clarity of the story, I have trouble with Figs. 5 and 6. First, you describe Fig. 6 in the text before you describe Fig. 5. That is an easy fix. Then when I look at cases in each of the figures the controls are x/y meaning X= control levels for Y embryos. But the X in the K02288 perturbed embryos = abnormal levels seen. Then, in the axitinib treated embryos the X is again meaning = controls. To the casual reader they could misinterpret the data to suggest almost no effects. It seems to me that in the scoring if you allowed X to always equal scored as a control, you would allow interpretation to be more easily weighed by the reader. Also, by what criterion is your decision on control vs perturbed decided. I understand those cases where relative to the staining of SM37 there is no staining in the perturbed, but what about your decision on where to score perturbed vs control in those where the staining appears reduced?

2. In the text for fig.2 you say first that Fig. 2 - 2B' reports the DMSO control and K02288 knockdown. You go on to say that Fig. 2B-B' shows Alk 1/2 Alk3/6 double knockdowns. Something is wrong - the same panel can't show two different knockdowns.

3. In the next paragraph you mislabel in the text the smad 158 panels - you say Fig. 2C when the figure shows that to be 2D-D'. Fig. 2C appears to be a vegetal view of the K02288 knockdown.

4. In that same paragraph, you suggest that continuous BMP is necessary for phase 2 to work. You did the continuous experiment but you didn't do the experiment you suggest - shorter pulses, so unless you do the shorter pulses you should soften this suggestion.

5. Fig. 3 puts me back into the numbering dilemma again. This time it is about the recurrent rod. You say that 27/32 show abnormalities in the text and 5/32 are at least close to normal. Yet, in Fig 3E-E', you use the 27/32 and point to the recurrent rod. C-C' doesn't indicate how many are normal vs abnormal. Since the denominator for the panel is 32 I assume that C-E are the same group of embryos. So, what am I seeing? First, an example of the common abnormal phenotypes seen in an unknown number of the 32, second, 22 that have the body rod abnormal, and 27 that have a recurrent rod abnormality. Then you label this figure as "-selectively affects body rod and recurrent rod development". Your figure doesn't show that selectivity - what about post-oral rods and anterolateral rods? And, based on concern #1, the reader is confused about what is X. Finally, your conclusion about primary effects is biased it seems to me - because the reader will say "wait a minute you show highly significant effects on body rods also, so how do you decide primary"? Why not just admit that an effect is an effect rather than making a judgement on a select (or favored) outcome.

Reviewer 2*Advance summary and potential significance to field*

See original review

Comments for the author

Thank you for addressing my concerns. I have no further requests for revisions.

Second revision

Author response to reviewers' comments

We thank the reviewers for taking the time to read our responses and closely analyzing our work to ensure that it is as clear as possible to readers. Below we have made note of the changes we have made in accordance with the reviewers' suggestions as well as our responses to their comments.

All changes in the manuscript text and figure legends are indicated by **blue font**.

Reviewer 1:

SUGGESTIONS TO AUTHORS

In reading the revised manuscript, some of the concerns have been corrected, especially concerns over the Figure legends. Below, however there are still issues of concordance. In reading the text and paying close attention to the concordance with the Figures, there are a number of issues. They are given below. These need to be fixed so the text matches what is shown in the figures.

1. For clarity of the story, I have trouble with Figs. 5 and 6. First, you describe Fig. 6 in the text before you describe Fig. 5. That is an easy fix. Then when I look at cases in each of the figures the controls are x/y meaning X= control levels for Y embryos. But the X in the K02288 perturbed embryos = abnormal levels seen. Then, in the axitinib treated embryos the X is again meaning = controls. To the casual reader they could misinterpret the data to suggest almost no effects. It seems to me that in the scoring if you allowed X to always equal scored as a control, you would allow interpretation to be more easily weighed by the reader. Also, by what criterion is your decision on control vs perturbed decided. I understand those cases where relative to the staining of SM37 there is no staining in the perturbed, but what about your decision on where to score perturbed vs control in those where the staining appears reduced?

The text has been adjusted to introduce figure 5 before figure 6. We thank the reviewer as this change has helped improve the overall flow of the story.

With regards to the display of X/Y scoring in figures 5 and 6, we feel that it makes the most sense to have the numbers in each panel reflect the number of embryos (X) out of total embryos scored (Y) that match the representative image shown. The figure legends have been adjusted to clarify this point as best as possible.

For our scoring criterion, most mRNAs (*gata3*, *scl*, *bmp2/4*, and *bmp5/8*) were scored for any detectable expression or complete absence of expression in the DS or RR-PMCs which were visualized with *sm37* or *vegr1G-10* as a second stain. The only mRNA that was scored differently was *sm29*. In control embryos, this mRNA is expressed throughout most of the PMC syncytium, but is clearly enriched in the DS and RR-PMCs by qualitative analysis. In perturbed embryos, the enrichment in the DS and RR-PMCs is no longer present and the expression of *sm29* where it appears in the syncytium is of uniform intensity. The main text and the text of the legends has been adjusted to clarify this point.

2. In the text for fig.2 you say first that Fig. 2 - 2B' reports the DMSO control and K02288 knockdown. You go on to say that Fig. 2B-B' shows Alk 1/2 Alk3/6 double knockdowns. Something is wrong - the same panel can't show two different knockdowns.

Here we are only comparing the morphology of our embryos treated with K02288 to the morphology of Alk1/2 + Alk3/6 double morphants as described in the study by Hailot et al., 2015. The text has been adjusted to clarify this point.

3. In the next paragraph you mislabel in the text the smad 158 panels - you say Fig. 2C when the figure shows that to be 2D-D'. Fig. 2C appears to be a vegetal view of the K02288 knockdown.

We thank the reviewer for pointing out this mistake and have corrected it.

4. In that same paragraph, you suggest that continuous BMP is necessary for phase 2 to work. You did the continuous experiment but you didn't do the experiment you suggest - shorter pulses, so unless you do the shorter pulses you should soften this suggestion.

We draw this conclusion based on the knowledge that by the time we are adding K02288 to the embryo cultures, pSmad has already accumulated in the dorsal PMCs and remains strongly activated in these cells throughout the rest of development in control embryos. Our observation that in K02288-treated embryos pSmad is no longer present at later stages indicates that any pSmad present before the drug was added has been cleared and no new pSmad been activated. As we did not perform a series of experiments adding the drug at later stages for shorter pulses and looking at pSmad activity, we refrained from making stronger conclusions and feel that saying our observations "suggest" the presented conclusion is appropriate.

5. Fig. 3 puts me back into the numbering dilemma again. This time it is about the recurrent rod. You say that 27/32 show abnormalities in the text and 5/32 are at least close to normal. Yet, in Fig 3E-E', you use the 27/32 and point to the recurrent rod. C-C' doesn't indicate how many are normal vs abnormal. Since the denominator for the panel is 32 I assume that C-E are the same group of embryos. So, what am I seeing? First, an example of the common abnormal phenotypes seen in an unknown number of the 32, second, 22 that have the body rod abnormal, and 27 that have a recurrent rod abnormality. Then you label this figure as "-selectively affects body rod and recurrent rod development". Your figure doesn't show that selectivity - what about post-oral rods and anterolateral rods? And, based on concern #1, the reader is confused about what is X. Finally, your conclusion about primary effects is biased it seems to me - because the reader will say "wait a minute you show highly significant effects on body rods also, so how do you decide primary"? Why not just admit that an effect is an effect rather than making a judgement on a select (or favored) outcome.

In the text, we state that 22/32 embryos showed abnormal body rods and 27/32 showed abnormal recurrent rods, which matches the figure. C & C' show a single representative K02288-treated embryo, which illustrates the typical perturbed morphology that was observed. D - E' are intended to further display the most striking effects of the drug treated that were observed. The figure legend has been adjusted to clarify that E & E' show a representative image of perturbed recurrent rods that lack the typical, distal bend.

We judge that K02288 "primarily" affects the body rod and recurrent rods based on the severity of the effect. As described in the text and the figure, the body rod in K02288-treated embryos is only on average ~30% of the length seen in DMSO-treated embryos. All other elements (including the recurrent rods) in K02288-treated embryos were at least 60% of the length seen in DMSO-treated embryos. The recurrent rods stand out separately as they were consistently missing a bend that arises during normal development. The length of the post-oral and anterolateral rods was only modestly reduced and they were otherwise normal morphologically.

All of these effects are described in the text and we present the BMP-signaling pathway as having a critical role in dorsal skeletal development due to the more striking morphological effects on the body rods and recurrent rods, as well as a striking, selective effect on dorsal PMC gene expression. We make note of the possible contribution of the pathway to skeletal development in other regions.

References

Haillot, E., Molina, M. D., Lapraz, F. & Lepage, T. 2015. The Maternal Maverick/GDF15-like TGF- β Ligand Panda Directs Dorsal-Ventral Axis Formation by Restricting Nodal Expression in the Sea Urchin Embryo. *PLoS Biol*, 13, e1002247.

Third decision letter

MS ID#: dev.205344R2

MS TITLE: BMP signaling regulates dorsal skeletal growth in the sea urchin embryo

AUTHORS: William Douglas and Charles A. Ettensohn

Dear Dr Ettensohn,

I am happy to tell you that your manuscript has been accepted for publication in Development, pending our standard publication integrity checks.